# Generalizations of Kitaev's honeycomb model from braided fusion categories

Luisa Eck

*Rudolf Peierls Centre for Theoretical Physics, University of Oxford,*
*Parks Road, Oxford, OX1 3PU, United Kingdom*

Paul Fendley

*Rudolf Peierls Centre for Theoretical Physics, University of Oxford,*
*Parks Road, Oxford, OX1 3PU, United Kingdom and*
*All Souls College, University of Oxford, Oxford, OX1 4AL, United Kingdom*
(Dated: August 9, 2024)

Fusion surface models, as introduced by Inamura and Ohmori [1], extend the concept of anyon chains to 2+1 dimensions, taking fusion 2-categories as their input. In this work, we construct and analyze fusion surface models on the honeycomb lattice built from braided fusion 1-categories. These models preserve mutually commuting plaquette operators and anomalous 1-form symmetries. Their Hamiltonian is chosen to mimic the structure of Kitaev's honeycomb model, which is unitarily equivalent to the Ising fusion surface model. In the anisotropic limit, where one coupling constant is dominant, the fusion surface models reduce to Levin-Wen string-nets. In the isotropic limit, they are described by weakly coupled anyon chains and are likely to realize chiral topological order. We focus on three specific examples: (i) Kitaev's honeycomb model with a perturbation breaking time-reversal symmetry that realizes chiral Ising topological order, (ii) a $\mathbb{Z}_N$ generalization proposed by Barkeshli *et al.* [2], which potentially realizes chiral parafermion topological order, and (iii) a novel Fibonacci honeycomb model featuring a non-invertible 1-form symmetry.

## CONTENTS

## I. INTRODUCTION

Kitaev's exactly solvable spin-$\frac{1}{2}$ model on the honeycomb lattice [3] displays a range of exotic quantum spin liquid phases, both topologically ordered and gapless. When time-reversal symmetry is broken, it supports non-abelian topological order with Ising anyons, which hold promise for fault-tolerant quantum computation [4]. This non-abelian phase also features gapless edge modes described by chiral Ising conformal field theories. The edge

modes, characteristic of chiral topological order, are of great interest to experiments. Recently their signatures were observed in the thermal Hall conductances of the Kitaev material candidate $\alpha$-RuCl$_3$ [5, 6].

Numerous generalizations of Kitaev's model have been developed independently, including extensions to higher spin [7–9] and to $\mathbb{Z}_N$ [2, 10, 11]. In this paper, we take a systematic approach to constructing generalizations of Kitaev's honeycomb model. We develop techniques introduced in [1] to show how braided fusion categories provide a convenient framework as well as tools to explore such topologically ordered phases.

A deep connection between lattice statistical-mechanical models and fusion categories predates the definition of the latter. Transfer matrices of 2d classical lattice models can be written in terms of the generators of algebras such as that of Temperley and Lieb, the very same algebras that underlie the construction of knot invariants like the Jones polynomial [12–14]. Fusion categories provide an elegant understanding of the common mathematical structure, a connection that became readily apparent in the "anyon chain" limit of these models [15, 16]. Such lattice models inherit a symmetry algebra from the input categories, resulting in non-invertible symmetries and dualities [17–19], meaning they cannot be implemented by unitary operators. Many of these lattice models, such as those of Andrews–Baxter–Forrester [20], have integrable limits [21] described by rational conformal field theories in the continuum.

Recently, Inamura and Ohmori [1] introduced a generalization to one dimension higher. Taking fusion 2-categories as input, their construction yields 3d classical and 2+1d quantum lattice models that naturally inherit symmetry. The latter, called fusion surface models, include Levin-Wen string-nets [22] in a special case. Strikingly, Kitaev's honeycomb model can be formulated as a fusion surface model. Moreover, we show how chiral topological order occurs in a fusion surface model, as a time-reversal-symmetry-breaking perturbation causing it can be realized in this framework.

We utilise this method to construct several models naturally generalizing the Kitaev honeycomb model. Our Hamiltonians contain non-commuting terms akin to those in Kitaev's model. By design, they possess mutually commuting local symmetries. The existence of such anomalous 1-form symmetries makes them promising candidates for various topologically ordered phases. Indeed, they reduce to Levin-Wen string-net models in a particular limit. Chiral topological order can occur because of complex phases in the Hamiltonian, and we provide evidence it does indeed occur.

In our work we take advantage of a simplification, in that many interesting cases do not require the general data of a 2-category, but rather only that of a braided fusion 1-category. Thus in essence generalizing the anyon-chain construction to 2+1d requires (at minimum) adding braiding to fusing. The ensuing models

typically seem to break time-reversal symmetry, and so provide candidates for chiral topological order without further modification.

We begin in Section II by reviewing how quantum chains constructed from fusion categories possess non-invertible symmetries. The 2+1d fusion surface models from braided fusion categories are described in Section III, where we show they become Levin-Wen models in a particular limit. In Section IV, we review how Kitaev's honeycomb model, including the magnetic-field perturbation and twist defects, is unitarily equivalent to the Ising fusion surface model. Section V investigates the $\mathbb{Z}_N$ generalization of Kitaev's honeycomb model constructed from the Tambara-Yamagami category, which turns out to be closely related to the model introduced in Barkeshli *et al.* [2]. We present further evidence that these models do indeed realize chiral topological order. Finally, a novel Fibonacci honeycomb model with a non-invertible 1-form symmetry is introduced in Section VI. Its time-reversal-symmetry breaking makes it a promising candidate for having chiral topological order.

## II. REVIEW OF QUANTUM ANYON CHAINS

Anyon chains are 1+1d quantum lattice models constructed from fusion categories [15, 16, 23]. They can be obtained from the anisotropic limit of the 2d classical statistical-mechanical models [12, 13, 17]. A key feature is that the Hamiltonian commutes with non-local "non-invertible" symmetries, whose generators are constructed from the fusion category. Such symmetries provide a natural generalization of Kramers-Wannier duality. More generally, these operators allow the construction of topological defects in the corresponding 2d classical lattice models, and so also yield topologically twisted boundary conditions in the 1d quantum chains. In this section, we review the construction of the anyon-chain Hamiltonian and its symmetries, laying the groundwork for the fusion surface model construction in Section III.

To construct the anyon chains, we start with the input fusion category $\mathcal{C}$, which consists of a finite number of simple objects $\{a, b, c, \dots\}$ with fusion rules

$$a \otimes b = \oplus_c N_{ab}^c \, c \, ,$$

where the $N_{ab}^c$ are non-negative integers. Fusion diagrams are planar trivalent graphs whose edges are labeled by objects in the category. At each trivalent vertex, the labels of its incident edges satisfy $N_{ab}^c \neq 0$. Evaluating a fusion diagram means associating an isotopy invariant to it. The diagram can be continuously deformed without changing its evaluation. Also, the evaluation is invariant under a set of manipulations described below in (2), (3). In this paper, we restrict to multiplicity-free fusion categories, meaning $N_{ab}^c = 0, 1$, and assume trivial Frobenius-Schur indicators for all objects. Except for Section V, we consider categories where all objects are self-dual, i.e. $0 \in a \otimes a$, where 0 is the identity object.

Self-duality implies that the lines in the fusion diagrams do not carry arrows. The fusion categories are also assumed to be unitary. The fusion diagrams then can be rotated at will, since any unitary fusion category admits a pivotal structure [3].

States in the anyon chain Hilbert space correspond to fusion trees of the form

$$|\{\Gamma_i\}\rangle = \begin{array}{c} \Gamma_1 \quad \Gamma_2 \quad \Gamma_3 \quad \Gamma_4 \quad \ldots \\ \hline \\ \rho \quad \rho \quad \rho \quad \rho \end{array}$$

Each vertical leg of the fusion tree is labeled by the same object $\rho \in \mathcal{C}$. The horizontal edges $\Gamma_i \in \mathcal{C}$ are the dynamical degrees of freedom. The local Hamiltonian $H_{i-1,i,i+1}$ acts on the fusion tree state as

$$H_{i-1,i,i+1}: \quad \begin{array}{c} \Gamma_{i-1} \quad \Gamma_i \quad \Gamma_{i+1} \\ \hline \\ \rho \quad \rho \end{array} \rightarrow \sum_h A_h \begin{array}{c} \Gamma_{i-1} \quad \Gamma_i \quad \Gamma_{i+1} \\ \hline \rho \; | \; \rho \\ \rho \; | \; h \; | \; \rho \end{array}, \tag{1}$$

with $h \in \mathcal{C}$ and constants $A_h \in \mathbb{R}$. Each term on the right-hand side of (1) can be evaluated using the F-moves of the fusion category,

$$\begin{array}{c} a \quad b \quad c \\ \diagdown \diagup \\ x \diagup \\ | \\ d \end{array} = \sum_y [F_d^{abc}]_{xy} \begin{array}{c} a \quad b \quad c \\ \diagdown \diagdown \diagup \\ \diagup y \\ | \\ d \end{array}, \tag{2}$$

together with the following identities to fuse lines to the fusion tree and remove bubbles:

$$\begin{array}{c} \underline{\quad\quad} \; a \\ \underline{\quad\quad} \; b \end{array} = \sum_c \sqrt{\frac{d_c}{d_a d_b}} \; \begin{array}{c} a \; \diagup\diagdown \; a \\ b \; \diagdown \, c \, \diagup \; b \end{array}, $$

$$c \; \diagup\!\!-\!\!\bigcirc\!\!-\!\!\diagdown \; a = \delta_{ac} \sqrt{\frac{d_b d_{b'}}{d_a}} \quad a \; \underline{\quad\quad} \; a. \tag{3}$$
$$\quad b'$$

Here $d_a$ denotes the quantum dimension of the object $a$. Explicitly, it follows that

$$\begin{array}{c} \Gamma_{i-1} \; \Gamma_i \; \Gamma_{i+1} \\ \hline \rho \, | \; \rho \\ \rho \; | \; h \; | \; \rho \end{array} = \sum_{\Gamma_i'} \sqrt{\frac{d_{\Gamma_i'}}{d_{\Gamma_i} d_h}} \begin{array}{c} \Gamma_{i-1} \; \Gamma_i' \; \Gamma_{i+1} \\ \hline \rho \diagup h \diagdown \rho \\ \rho \quad\quad \rho \end{array}$$

$$= \sum_{\Gamma_i'} \sqrt{d_h} [F_{\Gamma_{i-1}}^{\Gamma_i' h \rho}]_{\Gamma_i \rho} [F_{\Gamma_{i+1}}^{\rho h \Gamma_i}]_{\rho \Gamma_i'} \begin{array}{c} \Gamma_{i-1} \; \Gamma_i' \; \Gamma_{i+1} \\ \hline \\ \rho \quad \rho \end{array}.$$

By construction, the local Hamiltonian (1) commutes with topological lines labeled by objects $a \in \mathcal{C}$ acting

on the fusion tree from above:

$$\left[ \begin{array}{c} a \\ \overline{\Gamma_{i-1} \; \Gamma_i \; \Gamma_{i+1}} \\ \rho \, | \quad | \, \rho \\ \rho \; | \; h \; | \; \rho \end{array} , \begin{array}{c} a \\ \overline{\Gamma_{i-1} \; \Gamma_i \; \Gamma_{i+1}} \\ | \quad | \\ \rho \quad \rho \end{array} \right] = 0 \tag{4}$$

The action of the line $a$ on the fusion tree can be evaluated similarly as the action of the Hamiltonian,

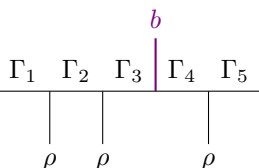

$$= \ldots [F_\rho^{\Gamma_{i-1}' a \Gamma_i}]_{\Gamma_{i-1} \Gamma_i'} [F_\rho^{\Gamma_i' a \Gamma_{i+1}}]_{\Gamma_i \Gamma_{i+1}'} \ldots$$

These topological lines therefore implement symmetries when they map the Hilbert space to itself, or dualities when they map to a different Hilbert space. Many examples are given in e.g. [17–19]. These symmetry generators obey the same fusion algebra as the corresponding object in the input category, and are non-invertible when $d_a > 1$. The local commutation relation (4) implies that any Hamiltonian $H = \sum_i C_i H_{i-1,i,i+1}$ with $C_i \in \mathbb{R}$ will commute with the topological line $a \in \mathcal{C}$. In particular, translation invariance is not required.

Twisted boundary conditions are implemented by gluing an additional vertical leg $b \in \mathcal{C}$ to the fusion tree:

$$\begin{array}{c} b \\ \Gamma_1 \; \Gamma_2 \; \Gamma_3 \, | \; \Gamma_4 \; \Gamma_5 \\ \hline \\ \rho \quad \rho \quad\quad \rho \end{array}$$

With twisted boundary conditions, it is possible to define a modified translation operator as a combination of the original translation operator and a unitary transformation. The unitary transformation is given by an F-move that moves the defect back to its original location [17].

## III. FUSION SURFACE MODELS FROM BRAIDED FUSION 1-CATEGORIES

Inamura and Ohmori [1] introduced *fusion surface models*, which naturally generalize anyon chains to 2+1 dimensions. Their construction uses fusion 2-categories as input. In this paper, we restrict to a simpler subclass of fusion 2-categories, namely braided fusion 1-categories. The resulting fusion surface models automatically preserve 1-form symmetries. While braiding is not a requirement for the 1+1d anyon chains, it is essential in the 2+1d case. We study a Hamiltonian that mirrors the structure of Kitaev's honeycomb model [3], a relation that will be reviewed in Section IV.

FIG. 1. Fusion surface model Hamiltonian.

## A. Construction and symmetries

As input fusion 2-category, we take the condensation completion $\mathrm{Mod}(\mathcal{B})$ of $\mathcal{B}$-module categories over a braided fusion 1-category $\mathcal{B}$. Practically speaking, this means the lattice construction relies only on the properties of $\mathcal{B}$ [1]. Throughout the paper, we assume $\mathcal{B}$ is multiplicity-free, as well as self-dual (except for Section V). To evaluate the resulting diagrams, the R-symbols are needed, namely

All fusion categories considered in this paper are unitary, so $(R_a^{bc})^{-1} = (R_a^{bc})^*$. In consequence, lines can be slid freely above and below fusion vertices.

States in the Hilbert space are fusion trees on the honeycomb lattice,

The black edges are labeled by objects $\Gamma_i \in \mathcal{B}$ and the four-valent vertices by morphisms $\Gamma_{ijk} \in \mathrm{Hom}(\Gamma_i \otimes \Gamma_j, \Gamma_k \otimes \rho)$. Following Inamura and Ohmori [1], the four-valent vertices are resolved into two trivalent vertices, at the expense of creating a new edge labeled by $\Gamma_{ijk} \in \mathcal{B}$:

(5)

All planar edges $\Gamma_i$ and $\Gamma_{ijk}$ on the surface of the fusion tree are dynamical degrees of freedom. As in the anyon chains, the vertical legs are fixed and labeled by the objects $\rho, \lambda \in \mathcal{B}$. In principle, $\rho$ and $\lambda$ can be different due to the bipartiteness of the honeycomb lattice, but in all examples discussed here, we choose $\lambda = \rho$. From now on, we use the graphical representation (5), where all edges are labeled by objects $\Gamma_i, \Gamma_{ijk}, \rho \in \mathcal{B}$. Unless stated otherwise, all vertical lines will be implicitly labeled by $\rho$.

We consider Hamiltonians of the form depicted in Fig. 1, reminiscent of Kitaev's honeycomb model [3]. We group the three types of operators around each plaquette $p$ into a single term $H_p$, so that $H = \sum_p H_p$. All coupling constants $J_x$, $J_y$, $J_z$ and weights $A_h$ are assumed to be real numbers. The Hamiltonian thus yields the simplest 2d analog of the anyon chain. The z-link term with coefficient $J_z$ is precisely the local anyon-chain Hamiltonian $H_{2i-1,2i,2i+1}$ from (1) and can be evaluated in the same way. However, because of the geometry of the honeycomb lattice, the $J_x$ and $J_y$ fusion diagrams in Fig. 1 are no longer planar diagrams, as the line labeled by $h$ passes underneath the other lines. The braiding therefore is necessary to define the fusion surface models.

The x-link term with coefficient $J_x$ in Fig. 1 can be evaluated as follows:

$$= \sum_{\Gamma'_{klm}, \Gamma'_l, \Gamma'_k, \Gamma'_{ijk}} [F_{\Gamma_m}^{\rho h \Gamma_{klm}}]_{\rho \Gamma'_{klm}} [F_{\Gamma_k}^{\Gamma'_{klm} h \Gamma_l}]_{\Gamma_{klm} \Gamma'_l} [F_{\Gamma'_{klm}}^{\Gamma_l h \Gamma_k}]_{\Gamma'_l \Gamma'_k}$$

$$\times [F_{\Gamma_j}^{\Gamma'_k h \Gamma_{ijk}}]_{\Gamma_k \Gamma'_{ijk}} [F_{\Gamma_i}^{\Gamma'_{ijk} h \rho}]_{\Gamma_{ijk} \rho} (R_{\Gamma'_l}^{\Gamma_l h})^{-1} \sqrt{d_h}$$

The y-link term with coefficient $J_y$ can be evaluated analogously to the x-link term. In fact, it is related to the x-link term by combined spatial mirror reflection symmetry $\mathcal{P}$ and complex conjugation $\mathcal{K}$:

$$\ldots = \mathcal{P}\mathcal{K} \left( \ldots \right) \mathcal{K}^\dagger \mathcal{P}^\dagger. \quad (6)$$

Complex conjugation is necessary to conjugate the braiding phase. The z-link term is invariant under both $\mathcal{P}$ and $\mathcal{K}$. Because the x-link and y-link terms are not real, the fusion surface Hamiltonian breaks time-reversal symmetry unless there exists a unitary matrix $U$ such that $U H_p U^\dagger = (H_p)^*$.

Another new aspect of the 2+1d models is the exis-

tence of conserved plaquette operators $B_p^{(b)}$, $b \in \mathcal{B}$ [1]:

$$B_p^{(b)} : \quad \quad \rightarrow$$

$$=$$

(7)

In the diagram above, the blue $b$-lines fused to the lattice can be removed as usual, using the F-symbols and R-symbols of the braided fusion category $\mathcal{B}$. We will no longer write out the evaluation explicitly. These plaquette operators commute with the Hamiltonian (1) and among themselves. They can be combined into projectors $B_p$ satisfying $B_p^2 = B_p$, where

$$B_p = \sum_{b \in \mathcal{B}} \frac{d_b}{D} B_p^{(b)}, \quad \text{with } D = \sqrt{\sum_b d_b^2}. \quad (8)$$

Furthermore, the fusion surface models are invariant under 1-form symmetries $a \in \mathcal{B}$ fused to the honeycomb lattice from above,

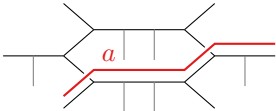

In order to commute with all terms in the Hamiltonian, the symmetry line $a$ must form a closed loop of any length, contractible or incontractible. Strictly speaking, 1-form symmetries have to act trivially on contractible loops; otherwise, the symmetry is more appropriately called a 1-symmetry [1, 24, 25]. Nonetheless we use the more common nomenclature of 1-form, even though a contractible loop occurs only with $B_p = 1$. The 1-form symmetry commutes with each individual term $H_p$, so any Hamiltonian $H = \sum_p C_p H_p$ with $C_p \in \mathbb{R}$ preserves it. An open string labeled by $a \in \mathcal{B}$ creates anyons at its endpoints when the ground state is gapped. Condensation defects [26–29] are networks of 1-form symmetry lines, thus these topological surfaces also commute with the Hamiltonian [1].

### B. Levin-Wen string-net limit

The phases of the fusion surface models (1) are constrained by their inherent 1-form symmetries [1]: A 1-form symmetry exhibits a 't Hooft anomaly when the associated anyons have nontrivial braiding [26, 30]. In the fusion surface model construction, the braiding of the 1-form symmetry generated by the object $a \in \mathcal{B}$ is characterized by the braiding phase $R_1^{a\bar{a}}$ of the input category $\mathcal{B}$, where $\bar{a}$ is the object dual to $a$. When this braiding phase is nontrivial, the 1-form symmetry line

$a$ is anomalous, requiring anomaly-matching. For invertible symmetries, the anomaly can be matched either by spontaneous symmetry breaking or by the phase being gapless. For non-invertible symmetries, generalized anomaly-matching conditions are explored in [30–32].

Spontaneous breaking of the 1-form symmetry $a$ results in topologically ordered ground states with anyonic excitations corresponding to $a$ [33]. Indeed, non-invertible symmetries in quantum chains can yield degenerate ground states and excitations exact up to exponentially small finite-size corrections, see e.g. [17, 19]. Mathematically, the modular tensor category describing the ensuing topological order takes the form $\mathcal{B}$ or $\mathcal{B} \boxtimes \mathcal{C}$, where $\mathcal{B}$ is the input category and $\mathcal{C}$ denotes another category describing emergent anyons [1, 34].

Two analytically tractable limits exist within the phase diagram, and are sketched in Fig. 2. Here we discuss one such limit, and in section III C the other.

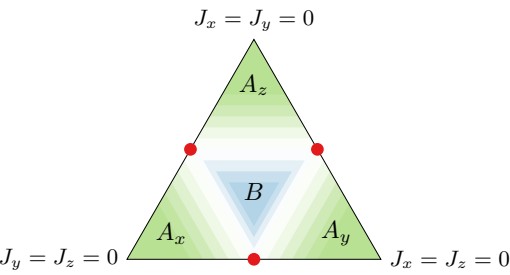

FIG. 2. Schematic phase diagram of the fusion surface model (1) with $J_x + J_y + J_z = 1$. The phases $A_x, A_y, A_z$ are characterized by non-chiral $\mathcal{Z}(\mathcal{B})$ topological order. At the red points, the model reduces to decoupled anyon chains, while in phase $B$, the chains are weakly coupled.

In the limit $J_z \rightarrow \infty$, the Hamiltonian (1) simplifies to a sum of commuting z-link terms that can be diagonalized independently. As mentioned before, the z-link term is equal to the local anyon chain Hamiltonian $H_{2i-1,2i,2i+1}$ acting on even sites (1). The ground states of the anyon chain in the completely staggered limit were computed in Section 8.1 of [17] for cases where $\mathcal{B}$ is either the $\mathbb{Z}_N$ Tambara-Yamagami category or the $\mathcal{A}_{k+1}$ category, and where $A_h = [F_\rho^{\rho\rho\rho}]_{0h}$ in (1) (all our examples satisfy these conditions up to a constant and an overall scaling). There exists one ground state for each object $r \in \mathcal{B}$, namely

$$|r\tilde{r}r\rangle = \quad \begin{matrix} r & \tilde{r} & r \\ | & | & | \\ \rho & \rho \end{matrix} \quad \text{with } |\tilde{r}\rangle = \frac{1}{\sqrt{d_r d_\rho}} \sum_x N_{r\rho}^x \sqrt{d_x} |x\rangle.$$

Consequently, the ground state subspace is effectively a honeycomb string-net,

$$\tilde{b} \ b \ \begin{matrix} c & \tilde{c} & c \\ & & \\ a & \tilde{a} & a \end{matrix} \ d \ \tilde{d} \quad \rightarrow \quad b \ \begin{matrix} c \\ \\ a \end{matrix} \ d$$

(9)

These states mix under the introduction of the x-link and y-link terms. We find that the lowest-order perturbation theory Hamiltonian in this subspace is the product of x-link and y-link terms around a plaquette:

$$
H^{\text{eff}} \sim \frac{J_x^2 J_y^2}{J_z^3} \sum_h A_h \quad \vcenter{\hbox{[diagram]}}
$$

$$
\to \frac{J_x^2 J_y^2}{J_z^3} \sum_h \tilde{A}_h \quad \vcenter{\hbox{[diagram]}}
\tag{10}
$$

In the first line of (10), we have to sum over the two resolutions of the four-valent blue vertices,

$$
\vcenter{\hbox{[diagram]}} = \vcenter{\hbox{[diagram]}} + \vcenter{\hbox{[diagram]}}
\tag{11}
$$

This large-$J_z$ result is straightforward to derive. At first order in perturbation theory, a single $J_x$ or $J_y$ link term changes two z-link states from their ground state $|r\tilde{r}r\rangle$ to an excited state,

$$
\vcenter{\hbox{[diagram]}} = \sum_{a',b',\tilde{a}',\tilde{b}',\dots} C_{a',b',\tilde{a}',\tilde{b}',\dots} \vcenter{\hbox{[diagram]}}
$$

Note that the x-link term changes $\tilde{a} \to \tilde{a}'$ as $\tilde{a}$ is not necessarily a simple object. The coefficients $C_{a',b',\dots}$ depend on the F-symbols and R-symbols. The overlap between this state and the original one can only be nonzero when $a' = a$, $b' = b$ and $d' = d$. In that case, the overlap reduces to

$$
\langle \tilde{a}' | \tilde{a} \rangle \langle \tilde{b}' | \tilde{b} \rangle \propto \left( \sum_x d_x [F_x^{rh\rho}]_{r\rho} \right)^2 .
$$

This follows from the expansion $|\tilde{a}\rangle \propto \sum_{x \in \mathcal{B}} N_{r\rho}^x \sqrt{d_x} |x\rangle$ and the action $[F_x^{rh\rho}]_{r\rho}$ of the Hamiltonian on each simple object $x$ in $\tilde{a}$. Hence, the overlap immediately vanishes if $N_{rr}^h = 0$, as it is the case for the Ising and $\mathbb{Z}_N$ Tambara-Yamagami examples discussed in Sections IV and V. Even when $N_{rr}^h \neq 0$, the overlap vanishes from the symmetry properties of the F-symbols we require (see e.g. [17]):

$$
\begin{aligned}
\sum_x d_x [F_x^{rh\rho}]_{r\rho} &= \sum_x d_x \sqrt{d_r d_\rho} \begin{bmatrix} r & h & r \\ \rho & x & \rho \end{bmatrix} \\
&= \sum_x d_x \sqrt{d_r d_\rho} \begin{bmatrix} r & r & x \\ \rho & \rho & h \end{bmatrix} \\
&= (d_r d_\rho) \sum_x d_x \begin{bmatrix} r & r & h \\ \rho & \rho & x \end{bmatrix} \begin{bmatrix} r & r & 0 \\ \rho & \rho & x \end{bmatrix} \\
&= (d_r d_\rho) \delta_{0h} N_{rr}^0 N_{\rho\rho}^0 = 0 \ \text{if } h \neq 0.
\end{aligned}
$$

In the first equality, we write the F-symbol in terms of the tetrahedral symbol, while in the second we employ the

column-permutation symmetry of the tetrahedral symbol, equation (2.42) in [17]. In the third equality, we use their (2.40) to insert a second tetrahedral symbol at the expense of a numerical factor, and in the fourth, we employ the orthogonality of the tetrahedral symbols in their (2.44). The overlap therefore vanishes for any Hamiltonian (as any term with $h = 0$ is simply a constant, we can simply set $A_0 = 0$ without loss of generality). This vanishing can be easily checked explicity for the Fibonacci fusion category with $h = r = \tau$ studied below.

At second order, the product of two adjacent $J_x$ and $J_y$ terms contains a part which acts diagonally on the z-link state that they share (due to $h$ being self-dual), even though they change the other two z-link states to orthogonal states. In a picture,

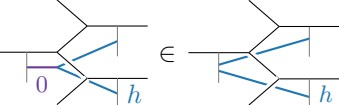

Hence, the lowest order effective Hamiltonian that preserves the ground state subspace arises at fourth order, and is the product of two $J_x$ and two $J_y$ terms around one plaquette, as depicted in the first line of (10). It contains a contribution that acts diagonally on the z-link states $|b\tilde{b}b\rangle$ and $|d\tilde{d}d\rangle$ on the left and right of the plaquette, and a contribution that flips the z-link states $|a\tilde{a}a\rangle$ and $|c\tilde{c}c\rangle$ on the top and bottom to different states $|a'\tilde{a}'a'\rangle$ and $|c'\tilde{c}'c'\rangle$ with the same energy. To show the last statement, we compute the overlap between the z-link acted upon by the $J_x$ and $J_y$ terms and the original z-link state. The new state is given by

$$
\vcenter{\hbox{[diagram]}} = \sum_{r',r'',x} \sqrt{d_x} N_{r\rho}^x d_h [F_x^{r''h\rho}]_{r\rho} [F_x^{\rho h r'}]^*_{\rho r} \vcenter{\hbox{[diagram]}}
$$

The overlap between the new state and another ground state $|r'\tilde{r}'r'\rangle$ is proportional to

$$
\sum_x d_x N_{r\rho}^x [F_x^{r'h\rho}]_{r\rho} [F_x^{\rho h r'}]^*_{\rho r} = \sum_x d_x N_{r\rho}^x \left| [F_x^{\rho h r'}]_{\rho r} \right|^2 > 0.
$$

The equality follows from the tetrahedral symmetry of the F-symbol. Hence, the ground states mix at fourth order in perturbation theory. In the ground state subspace, this fourth-order effective Hamiltonian thus acts as a Levin-Wen plaquette operator [22], as sketched in the second line of (10). The coefficient $\tilde{A}_h$ may be different from the coefficient $A_h$ in the first line because one of the two resolutions in (11) has an additional braiding phase depending on $h$. By construction, the Levin-Wen model realizes non-chiral topological order described by the Drinfeld centre $\mathcal{Z}(\mathcal{B})$ [22, 35]. Therefore the fusion surface model (1) also realizes such order in the $J_z \gg J_x, J_y$ limit (denoted as the $A_z$ phase in Fig. 2). We expect the same kind of topological order when either $J_x$ or $J_y$ dominate. In Appendix F, we discuss a more general commuting-projector Hamiltonian and its relation to the string-net models [22, 35–40].

### C.  Weakly coupled chains

When one coupling, e.g. $J_x$, is set to zero, the fusion surface model (1) reduces to a stack of $J_y$-$J_z$ chains with local Hamiltonian

$$-\sum_h A_h \left( J_y \;\; \substack{\Gamma_l \\ \Gamma_k \\ \Gamma_i \;\Gamma_{ijk}} \Gamma_{klm}\Gamma_m \quad + J_z \;\; \Gamma_{klm}\Gamma_m\Gamma_{mno} \right)$$

(12)

This Hamiltonian is diagonal in the $\Gamma_j$ and $\Gamma_l$ degrees of freedom. If they are all set to the identity object, the $J_y$-$J_z$ chain is precisely the anyon chain with the usual z-link Hamiltonian, cf. (1), and staggered couplings. The $\Gamma_l$ edges can be moved using F-symbols,

$$\substack{\Gamma_l \\ \Gamma_k \\ \Gamma_i\;\Gamma_{ijk}} \Gamma_{klm}\Gamma_m \;=\; \sum_{\Gamma'_{klm}} [F_\rho^{\Gamma_k\Gamma_l\Gamma_m}]_{\Gamma_{klm}\Gamma'_{klm}} \;\substack{\Gamma'_{klm}\;\;\Gamma_l}$$

(13)

Since the F-symbols are unitary in the lower two indices, moving the $\Gamma_l$ edge in this manner implements a unitary transformation of the $\Gamma_{klm}$ edge. Similarly, the $\Gamma_j$ edges can be shifted using a combination of F-symbols and R-symbols,

$$\substack{\Gamma_l \\ \Gamma_k \\ \Gamma_i\;\Gamma_{ijk}}\Gamma_{klm}\Gamma_m \;=\; \sum_{\Gamma'_{ijk}} [F_\rho^{\Gamma_i\Gamma_j\Gamma_k}]^{-1}_{\Gamma_{ijk}\Gamma'_{ijk}} R^{\Gamma_j\Gamma_{ijk}}_{\Gamma_k}(R^{\Gamma_j\Gamma_i}_{\Gamma'_{ijk}})^{-1}$$
$$\substack{\Gamma'_{ijk}\;\;\;\Gamma_l \\ \Gamma_j}$$

(14)

This transformation is also unitary because the R-symbols are unitary as well. By repeating these processes, all $\Gamma_j$ and $\Gamma_l$ edges can be moved to the same location, as illustrated below for a $J_y$-$J_z$ chain of size $L = 4$:

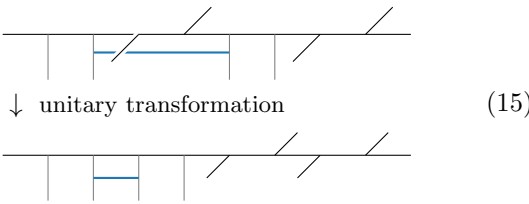

$$\downarrow \text{ unitary transformation} \qquad (15)$$

Except for the right-most term, the unitarily transformed Hamiltonian is exactly the anyon-chain Hamiltonian.

Having been moved together, the $\Gamma_j$ and $\Gamma_l$ edges then can be fused together, leading to a sum over objects at this location. For an open chain, taking the location to be the end amounts to a sum over boundary conditions on the anyon chain. The multiplicities in the sum lead to additional degeneracies in the spectrum for each boundary condition. For periodic boundary conditions, only one term in the Hamiltonian differs from the anyon chain.

This unitarily transformed Hamiltonian is effectively an anyon chain with a sum over twisted boundary conditions, again with multiplicities. We work out the unitary transformation (15) explicitly for the chain constructed from the Ising category in Appendix G. As seen there, the degeneracies grow exponentially with the size of the system. These large degeneracies can also be understood as arising from the remnants of plaquette operators $\bar{B}_p^{(b)}$ or of 1-form symmetries $W_\gamma^{(b)}$, acting as

$$\bar{B}_p^{(b)}: \quad \substack{b} \quad , \quad W_\gamma^{(b)}: \quad \substack{b}$$

Once the eigenvalues of the largest commuting set of these operators are fixed, the $J_z = 0$, $J_y = J_z$ model reduces to the anyon chain in the corresponding background fields. In many interesting cases including the examples we study, the continuum limit yields a conformal field theory.

Along the $J_y = J_z$, $J_x = 0$ line, the fusion surface model (1) is expected to be gapless and characterized by $L_y$ distinct 1d theories, which in the examples we study are CFTs. Upon introducing a small coupling $J_x \ll J_y = J_z$ between neighbouring critical chains, the fusion surface model realizes a coupled-wire system [41, 42]. When time-reversal symmetry is broken, chiral topological order is possible. We devote the remainder of the paper to discussing multiple example of such.

## IV.   KITAEV'S HONEYCOMB MODEL

The simplest non-trivial example of a fusion surface model of the form (1) is built from the Ising category, and its Hamiltonian is unitarily equivalent to the well-known Kitaev honeycomb model. This model has already been discussed in Section 5.2 in [1], but we will review and expand on it here to set the stage for its generalizations in Sections V and VI. Under a magnetic-field perturbation, Kitaev's honeycomb model is known to exhibit chiral Ising topological order [3]. By representing a related perturbation graphically we demonstrate explicitly that fusion surface models realize chiral topological order, as anticipated but not proven in [1].

### A.   Constructing Kitaev's honeycomb model from the Ising category

Kitaev [3] proposed an exactly solvable model of qubits on the vertices of a honeycomb lattice, with interactions between adjacent qubits depending on the direction of the connecting link, see Fig. 3. The Hamiltonian is

$$H^{\text{Kitaev}} = -J_x \sum_{a,b\in \text{x-link}} X_a X_b - J_y \sum_{a,c\in \text{y-link}} Y_a Y_c$$
$$-J_z \sum_{b,d\in \text{z-link}} Z_b Z_d,$$

(16)

where $X$, $Y$ and $Z$ denote the Pauli matrices. Conserved plaquette operators commute with the Hamiltonian (16) and among themselves,

$$B_p = Y_1 Z_2 X_3 Y_4 Z_5 X_6. \tag{17}$$

The physics of Kitaev's honeycomb model is well understood, as it can be mapped to free fermions when all plaquette operators are fixed to $B_p = \pm 1$ [3]. Its phase diagram is depicted in Fig. 4. In the anisotropic coupling limits, the effective Hamiltonian in perturbation theory reduces to the toric-code Hamiltonian and thus realizes doubled $\mathbb{Z}_2$ topological order in the phases $A_x$, $A_y$ and $A_z$. The phase $B$ near the isotropic point is gapless.

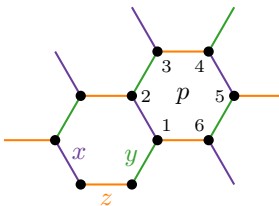

FIG. 3. Kitaev's honeycomb model with qubits on the vertices and interactions depending on the direction of the link

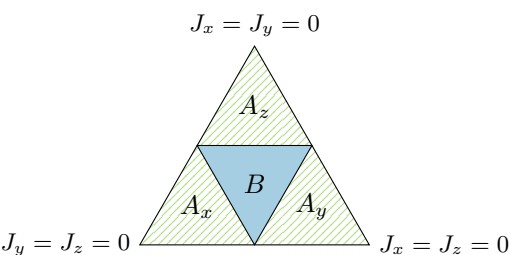

FIG. 4. Phase diagram of Kitaev' honeycomb model with $J_x + J_y + J_z = 1$, exhibiting gapped phases $A_x$, $A_y$, $A_z$ and a gapless phase $B$ [3]

Remarkably, Kitaev's honeycomb model is unitarily equivalent to the fusion surface model built from the Ising category [1]. The Ising category consists of three objects $\{0, 1, \sigma\}$ with the identity object denoted as 0. The non-abelian object $\sigma$ has $d_\sigma = \sqrt{2}$ and obeys the fusion rules $\sigma \otimes \sigma = 0 \oplus 1$ and $\sigma \otimes 1 = \sigma$. In the fusion surface model construction, we pick $\rho = \sigma$ so that all vertical legs of the fusion tree are labeled by $\sigma$. Half of the planar edges $\Gamma_i$ are also labeled by $\sigma$, with the remaining planar edges representing the dynamical degrees of freedom $\Gamma_{ijk} \in \{0, 1\}$ of the quantum state:

$$|\{\Gamma_{ijk}\}\rangle = \text{(diagram)} \tag{18}$$

In this and all subsequent fusion diagrams, the thin black lines are labeled by $\sigma$, and the red dotted lines are labeled

by $\{0, 1\}$. Consequently, the Hilbert space is spanned by states of qubits on a honeycomb lattice. The action of the local Hamiltonian on the states (18) is given by

$$H_p : \ -\left( J_x \text{(diagram)} + J_y \text{(diagram)} + J_z \text{(diagram)} \right) \tag{19}$$

Because of the fusion rule $\sigma \otimes 1 = \sigma$, the Hamiltonian does not change the $\sigma$ labels on half of the planar edges, consistent with these labels being fixed initially.

The x-link, y-link and z-link terms in the local Hamiltonian (19) can be evaluated as discussed in Section III A, with detailed calculations provided in Appendix A. The z-link term evaluates to $Z_{klm}Z_{mpq}$ just as half of the terms in the Ising chain, and the x-link and y-link terms yield $-Y_{klm}X_{ijk}$ and $Y_{ijk}X_{jno}$ respectively. Thus, the full fusion surface Hamiltonian $H = \sum_p H_p$ with $H_p$ as defined in (19) is equal to

$$H = - J_x \sum_{b,a \in \text{x-link}} (-Y_b X_a) - J_y \sum_{a,c \in \text{y-link}} Y_a X_c$$
$$- J_z \sum_{b,d \in \text{z-link}} Z_b Z_d. \tag{20}$$

After a unitary rotation $e^{i\pi Z/4}$ of all qubits on one sublattice of the bipartite honeycomb lattice, the Hamiltonian (20) becomes Kitaev's honeycomb model (16).

By construction, the fusion surface model (20) has conserved plaquette operators $B_p^{(1)}$ as defined in (7),

$$\text{(diagram)} = \text{(diagram)}. \tag{21}$$

In operator form, (21) yields

$$B_p^{(1)} = -X_{klm} Z_{ijk} Y_{jno} Y_{ors} Z_{prt} X_{mpq}.$$

Equivalently, $B_p^{(1)}$ is the product of all terms in the Hamiltonian around the plaquette. After the unitary rotation described above, this is precisely the conserved plaquette operator (17) of Kitaev's honeycomb model.

The fusion surface model (20) is also invariant under a $\mathbb{Z}_2$ 1-form symmetry,

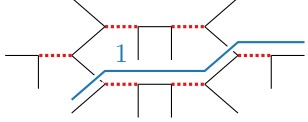

In operator form, this is the product of the terms in the Hamiltonian along the path, here leading to alternating

$X$ and $Y$ matrices,

$$\ldots Y_{klm} X_{mpq} Y_{prt} \ldots$$

It follows immediately that the 1-form symmetry is fermionic as it inherits the braiding phase $R_0^{11} = -1$ of the input category. Open strings of this 1-form symmetry create fermionic $\mathbb{Z}_2$ anyons at their endpoints when the ground state is gapped.

For general fusion surface models, we showed in Section III B that they reduce to a Levin-Wen string-net in the large $J_z$ limit. The Ising fusion surface model discussed here reduces in fact to the $\mathbb{Z}_2$ toric code because its Hamiltonian (20) does not feature the $\sigma$-line. More explicitly, as $J_z \to \infty$, there are two ground states $Z_a = Z_b = \pm 1$ on each z-link. The lowest order Hamiltonian acting in this ground-state subspace is the following,

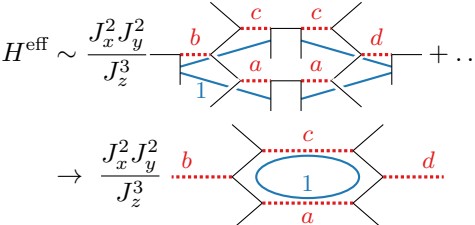

This effective Hamiltonian does not change the location of the qubits on the string-net, as it does not contain the $\sigma$-loop, and is therefore equivalent to the toric code rather than the Ising string-net.

## B. Chiral Ising topological order in Kitaev's honeycomb model perturbed by a magnetic field

The phase $B$ in the center of the phase diagram in Fig. 4 is gapless but becomes chiral Ising topological order once time-reversal symmetry is broken [3, 43]. On a torus or infinite cylinder, the system is gapped with three ground states corresponding to the objects in the Ising category. Gapless edge modes occur for open boundaries, and on an infinitely long strip, the system becomes gapless and chiral Ising CFTs propagate on the top and bottom edges. Time-reversal symmetry can be broken explicitly by adding a magnetic field perturbation $V$ to Kitaev's honeycomb Hamiltonian (16), given by

$$V = -\sum_j (h_x X_j + h_y Y_j + h_z Z_j).$$

This perturbation $V$ does not commute with the conserved plaquette operators (17). In perturbation theory, the lowest-order effective perturbation that commutes with the plaquette operators is [3]

$$V_{\text{eff}}^{(3)} \sim \frac{h_x h_y h_z}{J^2} \sum_{j,k,l} X_j Y_k Z_l. \tag{22}$$

The effective perturbation $V_{\text{eff}}^{(3)}$ consists of products of adjacent link terms in the Hamiltonian, which necessarily can be represented in the fusion surface model framework. Kitaev [3] computed the spectrum of the Hamiltonian with the time-reversal symmetry breaking $V^{(3)}$ perturbation explicitly using free-fermion methods and showed that it exhibits chiral Ising topological order. Thus, the Ising fusion surface model (19) with the perturbation (22) serves as an example of a fusion surface model with chiral topological order. We thus confirm that fusion surface models can exhibit chiral topological order, as anticipated in [1].

## C. Twist defects

Another interesting point is the interpretation of the topological $\sigma$-line fused to the honeycomb lattice from above. The $\sigma$-line does not act as a 1-form symmetry because it changes the location of the qubits on the honeycomb fusion tree. Still, a closed $\sigma$-loop of any length commutes with the Hamiltonian and can thus be interpreted as a 1-form duality. For example, fusing a $\sigma$-loop to one plaquette maps Kitaev's honeycomb model to a model with the following modified plaquette term:

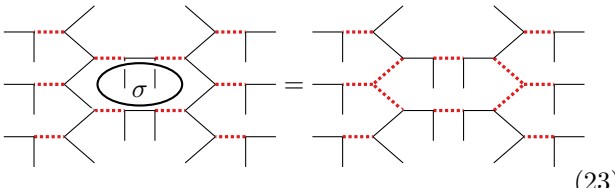

$$(23)$$

The lattice of qubits is different in the right picture, and also the terms in the Hamiltonian change. For instance, the z-link term changes from a $Z_{klm} Z_{mpq}$ interaction in to an $X_m$ interaction, reminiscent of the Kramers-Wannier duality in the Ising chain. This 1-form duality $D_\sigma$ is non-invertible as it obeys the same fusion algebra $D_\sigma^2 = \mathbb{I} + D_1$ as the $\sigma$-object, with $D_1$ denoting the $\mathbb{Z}_2$ 1-form symmetry. The fusion relation above implies that the 1-form duality does not implement a simple one-to-one mapping of the energy spectrum, as it annihilates states in the $D_1 = -1$ sector, similar as non-invertible dualities in 1+1 dimensions [17–19].

Open $\sigma$-strings create twist defects instead of anyons. Twist defects, introduced in the context of the toric code by Bombin [44], are located at the endpoints of lattice dislocations. Petrova $et$ $al.$ [45, 46] explored lattice dislocations and twist defects in the gapped phase of Kitaev's honeycomb model. Here, we review the key results from Petrova $et$ $al.$ [46] to compare them with the action of the open $\sigma$-string in the fusion surface model.

In the $J_z \gg J_x, J_y$ phase of Kitaev's honeycomb model, the ground state is in the $B_p^{(1)} = +1$ sector, with low-energy excitations corresponding to flipped plaquettes $B_p^{(1)} = -1$. These $\mathbb{Z}_2$ vortex excitations come in two flavors, $e$ and $m$, which live on alternating rows of the

honeycomb lattice:

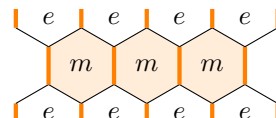

Here the strong $z$-bonds are represented by the thick orange lines and the weak $x$- and $y$-bonds by the thin black lines. At low energies, only vortices in the same row can be created or annihilated pairwise. They can move within their row or hop to the next-nearest row of the same vortex type. The creation of $f = e \otimes m$ anyons is effectively forbidden at low energies. When the number of rows is odd, $e$ and $m$ plaquettes cannot be consistently defined, reducing the ground state degeneracy on a torus from four to two.

8-2 lattice dislocations are created by removing certain link terms in the Hamiltonian, leading to defect sites involved in only two link terms instead of three:

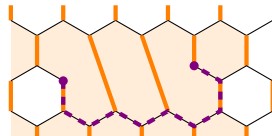

The two twist defect sites, which lack one weak bond each, are indicated by violet circles. The dashed line represents the branch cut, which disrupts the vortex flavor pattern. Its position is a gauge choice, hence only the defect sites at its endpoints are physical. Each pair of dislocations encodes one nonlocal qubit, which increases the ground state degeneracy by a factor of two. For $n \geq 1$ dislocation pairs on a torus with an even number of rows, the ground state degeneracy increases to $2^{n+1}$ (the first dislocation pair does not affect the ground state degeneracy because the branch cut renders the $e$ and $m$ flavors indistinguishable). This demonstrates that the quantum dimension of the dislocation defect is $\sqrt{2}$, consistent with the $\sigma$ object in the Ising category. The twist defects are distinct from intrinsic anyons, which are excited states of the Hamiltonian [47]. However, twist defects can also be leveraged for topological quantum computation, employing measurement-based braiding approaches [48].

In the Ising fusion surface model (19), fusing an open $\sigma$-string to the lattice relocates the qubits along its path and creates twist defects at the endpoints:

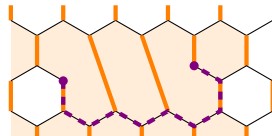

$$(24)$$

The twist defects are the thick black lines added to the fusion tree, and by definition they obey the same fusion and braiding rules as the Ising anyon. The $\sigma$-string is topological away from its endpoints, similar as the branch cut line. The difference to the lattice dislocations studied in [45, 46] is that fusing the $\sigma$-string to the lattice not only alters the positions of the terms in the Hamiltonian but also modifies their operator form, as discussed above in (23).

## V. $\mathbb{Z}_N$ GENERALIZATION OF KITAEV'S HONEYCOMB MODEL

Starting from the $\mathbb{Z}_N$ Tambara-Yamagami category for odd $N > 2$, we build a $\mathbb{Z}_N$ symmetric fusion surface model generalizing Kitaev's honeycomb model. This fusion surface model turns out to be closely related to the $\mathbb{Z}_N$ generalization proposed by Barkeshli *et al.* [2]. A coupled-wire analysis suggests chiral parafermion topological order in the $\mathbb{Z}_3$ model with additional and appropriately tuned interactions [2]. Our numerical studies of the entanglement spectrum indicate that this chiral parafermion topological order seems to persist even when the interactions are not fine-tuned.

### A. Constructing the Hamiltonian from the G-crossed braided $TY(\mathbb{Z}_N)$ category with odd $N$

The $\mathbb{Z}_N$ Tambara-Yamagami fusion categories [49] are generalizations of the Ising category ($N = 2$) to categories with $N$ abelian objects. We assume odd $N$ here to use the F-symbols and R-symbols found in Section XI.G.2 in [50]. The abelian objects are labeled by integers $h$ modulo $N$, and their fusion rules are the group multiplication rules of $\mathbb{Z}_N$,

$$h \otimes g = [h + g]_N \quad \forall h, g \in \{0, 1, \ldots, N - 1\},$$

with addition modulo $N$ on the right hand side. This category also contains a non-abelian object $\sigma$ with quantum dimension $d_\sigma = \sqrt{N}$ and fusion rules

$$\sigma \otimes h = h \otimes \sigma = \sigma, \quad \sigma \otimes \sigma = \bigoplus_{h=0}^{N-1} h.$$

While $\sigma$ is always self-dual, the abelian objects are no longer self-dual for $N > 2$, and so their lines in the fusion diagrams carry arrows. Charge conjugation acts on the abelian objects as $h \to h^{-1} = N - h$, i.e. reverses their direction.

The crucial difference between the Ising category and the $\mathbb{Z}_N$ Tambara-Yamagami category with $N > 2$ is that the latter only admits $G$-crossed braiding [51, 52]. The Tambara-Yamagami category is a $G$-graded fusion category $\mathcal{C}_G = \mathcal{C}_\mathbf{0} \oplus \mathcal{C}_\mathbf{1}$, with $G$ being the $\mathbb{Z}_2$ charge conjugation symmetry. The grading structure is respected by the fusion rules, i.e. $a_\mathbf{g} \otimes b_\mathbf{h} = \oplus_{c_\mathbf{gh}} N_{ab}^c c_\mathbf{gh}$ for $g, h \in G$. The graded component $\mathcal{C}_\mathbf{0} = \mathbb{Z}_N^{(r)}$ contains the abelian objects. Their braiding depends on an integer parameter

$r = 1, \ldots, N-1,$

$$\underset{a}{\overset{b \quad c}{\curlyvee}} = R_a^{bc} \;\; \underset{a}{\overset{b \quad c}{\vee}} \;, \qquad R_{[a+b]_N}^{ab} = e^{i\frac{2\pi}{N}rab} \;\; \text{for } a,b \in \mathbb{Z}_N.$$

The topological twist factors of the abelian objects are

$$\theta_a \equiv (R_1^{aa^{-1}})^{-1} = e^{i\frac{2\pi}{N}ra^2} \text{ for } a \in \mathbb{Z}_N. \tag{25}$$

The other graded component $\mathcal{C}_1$ of the Tambara-Yamagami category contains the non-abelian object $\sigma$. In the graphical calculus, $G$-crossed braiding between the $\sigma$ object and the abelian objects can be depicted as [50]

$$R^{\sigma h} = \underset{\sigma \quad h}{\overset{h^{-1} \quad \sigma}{\diagdown\!\!\!\!\diagup}} \;, \qquad R^{h\sigma} = \underset{h \quad \sigma}{\overset{\sigma \quad h}{\diagup\!\!\!\!\diagdown}}$$

The $\sigma$-line thus applies the charge-conjugation-group action to the abelian object $h$ when it crosses over their wordline. Conversely, when the $\sigma$-line undercrosses the $h$-line, nothing happens to the $h$-line. The R-symbols involving $\sigma$ are given by [50]

$$R_\sigma^{\sigma a} = R_\sigma^{a\sigma} = (-1)^{ra} e^{\frac{-i\pi r}{N}a^2}.$$

For oriented lines, the F-symbols are defined as

$$\underset{d}{\overset{a \quad b \quad c}{\underset{x}{\curlyvee\!\!\!\curlyvee}}} = \sum_y [F_d^{abc}]_{xy} \underset{d}{\overset{a \quad b \quad c}{\curlyvee\!\!\!\underset{y}{\curlyvee}}}$$

The non-trivial F-symbols of the Tambara-Yamagami category involve $\sigma$ and are given by

$$[F_\sigma^{a\sigma b}]_{\sigma\sigma} = [F_b^{\sigma a\sigma}]_{\sigma\sigma} = e^{\frac{2\pi ir}{N}ab}, \; [F_\sigma^{\sigma\sigma\sigma}]_{ab} = \frac{1}{\sqrt{N}}e^{\frac{-2\pi ir}{N}ab}.$$

The local $\mathbb{Z}_N$ fusion surface Hamiltonian $H_p$ acts as

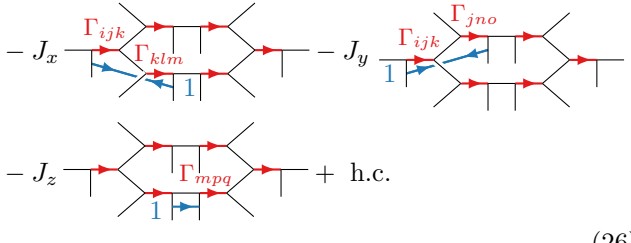

$$\tag{26}$$

The dynamical degrees of freedom on the fusion tree are now $N$-state qudits $\Gamma_{ijk} \in \{0, 1, \ldots, N-1\}$, denoted by red directed lines. When the blue line labeled by the 1-object undercrosses the $\sigma$-edge in the x-link and y-link term in (26), it changes its direction due to the $G$-crossed braiding. Apart from this important difference, the evaluation of the fusion diagrams (26) closely follows the calculation in Section IV and is detailed in Appendix B.

The z-link term evaluates to $Z_{klm}^{r\dagger}Z_{mpq}^r$, and the x-link and y-link terms to $X_{ijk}Z_{klm}^r X_{klm}^\dagger$ and $Z_{ijk}^r X_{ijk}X_{jno}^\dagger$ respectively, each multiplied by additional complex phases. Here $Z$ and $X$ are the $\mathbb{Z}_N$ clock and shift operators satisfying $XZ = \omega ZX$, with $\omega = e^{\frac{2\pi i}{N}}$. Explicitly,

$$Z = \begin{pmatrix} 1 & 0 & \cdots & & 0 \\ 0 & \omega & \cdots & & 0 \\ \vdots & \vdots & \ddots & & \vdots \\ 0 & 0 & \cdots & & \omega^{N-1} \end{pmatrix}, \quad X = \begin{pmatrix} 0 & 1 & 0 & \cdots & 0 \\ 0 & 0 & 1 & \cdots & 0 \\ \vdots & \vdots & \vdots & \ddots & \vdots \\ 1 & 0 & 0 & \cdots & 0 \end{pmatrix}.$$

The resulting $\mathbb{Z}_N$ fusion surface Hamiltonian is thus:

$$\begin{aligned} H = &- J_x \sum_{b,a\in\text{x-link}} (-1)^{rN}e^{-\frac{i\pi r}{N}} X_a Z_b^r X_b^\dagger \\ &- J_y \sum_{a,c\in\text{y-link}} (-1)^{rN}e^{\frac{i\pi r}{N}} Z_a^r X_a X_c^\dagger \\ &- J_z \sum_{b,d\in\text{z-link}} Z_b^{r\dagger} Z_d^r + \text{h.c.} \end{aligned} \tag{27}$$

The complex phases in the x-link and y-link terms (27) are such that the Hamiltonian is invariant under unitary charge conjugation $C$, acting as

$$CXC^\dagger = X^\dagger, \; CZC^\dagger = Z^\dagger, \; C(XZ)C^\dagger = X^\dagger Z^\dagger,$$

$$\text{with } C = \begin{pmatrix} 1 & 0 & 0 \\ 0 & 0 & 1 \\ 0 & 1 & 0 \end{pmatrix}. \tag{28}$$

Graphically, the charge conjugated terms are obtained by reversing the direction of the interaction (blue) lines in (26), and are precisely the hermitian conjugate terms. After a unitary transformation discussed in Appendix B, (27) becomes

$$\begin{aligned} H = &- J_x \sum_{b,a\in\text{x-link}} X_a X_b - J_y \sum_{a,c\in\text{y-link}} \omega^r Z_a^r X_a Z_c^r X_c \\ &- J_z \sum_{b,d\in\text{z-link}} Z_b^r Z_d^r + \text{h.c.} \end{aligned} \tag{29}$$

The $r=1$ and $r=N-1$ Hamiltonians have the same spectrum, as they are related by complex conjugation.

The $r = N-1$ case of (29) is closely related to the Hamiltonian

$$\begin{aligned} H^{\mathbb{Z}_N} = &- J_x \sum_{b,a\in\text{x-link}} X_a X_b - J_y \sum_{a,c\in\text{y-link}} (X_a Z_a^\dagger)(X_c Z_c^\dagger) \\ &- J_z \sum_{b,d\in\text{z-link}} Z_b Z_d + \text{h.c.} \end{aligned} \tag{30}$$

proposed in [2] as a $\mathbb{Z}_N$ generalization of Kitaev's honeycomb model. The only distinction is the complex phase $\omega^r$ in the y-link term guaranteeing charge conjugation invariance. The resulting finite-size spectra for $N=3$ are slightly different. However, with DMRG on an infinite cylinder, their energy and entanglement spectra agree.

Therefore we expect the $N=3$ models (30) and (29) to exhibit the same topologically ordered phases.

The $r = 1, N - 1$ fusion surface models (29) and the model (30) break time-reversal symmetry explicitly [2], as there is no unitary matrix $U$ such that $UHU^\dagger = H^*$. For such a unitary $U$ to exist for arbitrary coupling constants, it would need to map $Z \to Z^\dagger$ without changing $X$. However, such a mapping would change the commutation relations between $X$ and $Z$, making it impossible to implement by any unitary matrix.

### B. Anomalous $\mathbb{Z}_N$ 1-form symmetry

Plaquette operators $B_p^{(1)}$ that commute with the fusion surface model Hamiltonian (27) and among themselves are generated by fusing a 1-loop to the inside of a plaquette,

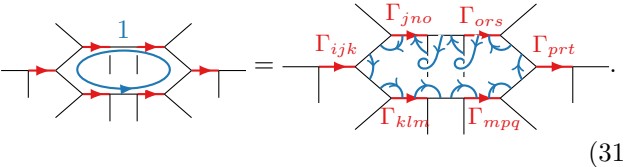

$$(31)$$

Note that the loop (blue) 1-line does not change its direction as it overcrosses the $\sigma$-edges. When the Tambara-Yamagami braiding parameter is set to $r = 1$, the plaquette operator is

$$B_p^{(1)} = \omega X_{klm} Z_{ijk} (ZX^\dagger)_{jno} (X^\dagger Z^\dagger)_{ors} Z_{prt}^\dagger X_{mpq},$$

which is the product of the terms in the Hamiltonian (26) around the plaquette (with the correct chiralities).

Similarly, the $\mathbb{Z}_N$ 1-form symmetry follows from fusing loops labeled by abelian objects to the honeycomb lattice from above. This symmetry is anomalous, meaning that endpoints of open strings have nontrivial exchange statistics. The exchange statistics for the model (30) were computed explicitly in [10, 53, 54], using the method described in [55]. The results are different, with $\theta_1 = \omega^2$ in [10] and $\theta_1 = \omega$ in [53, 54], as the models are slightly different, with the y-link term containing $XZ$ in the former but $XZ^\dagger$ in the latter. The fusion-surface-model construction directly yields the exchange statistics factor of the 1-form symmetry line $a$ to be the topological twist factor $\theta_a = \omega^{ra^2}$ (25) of the input category. This nontrivial statistics factor signals a 't Hooft anomaly, and anomaly matching requires the $\mathbb{Z}_N^{(r)}$ 1-form symmetry to be spontaneously broken or the phase to be gapless (as discussed in Section III B). When the 1-form symmetry is broken, the ground state is topologically ordered and its excitations include $\mathbb{Z}_N^{(r)}$ anyons.

Similar to the $\mathbb{Z}_2$ honeycomb model (cf. Section IV C), the $\sigma$-line fused to the honeycomb lattice from above does not give rise to a 1-form symmetry. Instead, it generates a topological twist defect line (when it is open) or a non-invertible 1-form duality (when it is closed).

For example, when a $\sigma$-loop is fused to one plaquette as depicted in (23), the z-link and y-link terms on this plaquette are modified to $Z_{klm}Z_{mpq}^\dagger \to X_m$ and $Z_{mpq}X_{mpq}X_{prt}^\dagger \to Z_m^\dagger X_p Z_r X_{prt}^\dagger$ respectively (for $r = 1$).

### C. Weakly coupled chains limit of the $\mathbb{Z}_3$ honeycomb model

The phase diagram of (30) for $N = 3$ was studied in [2] and more recently numerically in [10]. Its general structure is believed to be similar to the phase diagram of the $\mathbb{Z}_2$ model in Fig. 4. In the anisotropic limits, the effective Hamiltonian in perturbation theory is the $\mathbb{Z}_3$ toric code, and the model is characterized by doubled $\mathbb{Z}_3$ topological order [2]. The nature of the phase near the isotropic point $J_x = J_y = J_z$ is not yet fully established. Since the $\mathbb{Z}_3$ honeycomb Hamiltonian (30) breaks time-reversal symmetry, chiral topological order is possible. Indeed, numerics strongly indicate that the phase is gapped in the bulk but has chiral gapless edge modes [10], and that the ground state breaks time-reversal symmetry [56].

To gain a better understanding of this phase, we rephrase the coupled-wire analysis in [2] in the fusion surface model picture. When $J_x$ is set to zero, the $\mathbb{Z}_3$ honeycomb Hamiltonian reduces to decoupled $J_y$-$J_z$ chains, which are unitarily related to the anyon chain with twisted boundary conditions, cf. Section III B. The anyon chain built from the $\mathbb{Z}_3$ Tambara-Yamagami category is the $\mathbb{Z}_3$ Potts chain:

$$H^{\text{Potts}} = -\sum_j \left( h \; \overset{\Gamma_j \quad \Gamma_{j+1}}{\longrightarrow} \; + J \; \overset{\Gamma_j}{\longrightarrow} \; \right) + \text{h.c.}$$

$$= -h \sum_j Z_j^\dagger Z_{j+1} - J \sum_j X_j + \text{h.c.}$$

When $J_y = J_z$, the $J_z$-$J_y$ chain is critical and described by the Potts CFT [2]. The $J_x$ term which couples neighbouring chains can be rewritten in terms of left and right lattice parafermion operators $\hat{\alpha}_{L,i}$ and $\hat{\alpha}_{R,i}$. In the Potts chain, the lattice parafermions can be visualized as

$$\hat{\alpha}_{R,2j-1} = \overset{\Gamma_j}{\longrightarrow} = \left( \prod_{k=1}^{j-1} X_k \right) \omega Z_j$$

$$\hat{\alpha}_{R,2j} = \overset{\Gamma_j}{\longrightarrow} = \left( \prod_{k=1}^{j-1} X_k \right) X_j Z_j$$

$$\hat{\alpha}_{L,2j-1}^\dagger = \overset{\Gamma_j}{\longrightarrow} = \left( \prod_{k=1}^{j-1} X_k \right) \omega^2 Z_j^\dagger$$

$$\hat{\alpha}_{L,2j}^\dagger = \overset{\Gamma_j}{\longrightarrow} = \left( \prod_{k=1}^{j-1} X_k \right) X_j Z_j^\dagger.$$

$$(32)$$

Apart from an overall complex phase, these definitions agree with those in [2]. The lattice parafermions commute with the Hamiltonian away from their endpoints

and are discretely holomorphic current operators [21, 57–59]. Because the $J_z$-$J_y$ chains can be mapped to Potts chains, they also contain lattice parafermions with a similar visualization. The $J_x$ coupling, in terms of the lattice parafermions of the $J_z$-$J_y$ chains, is given by

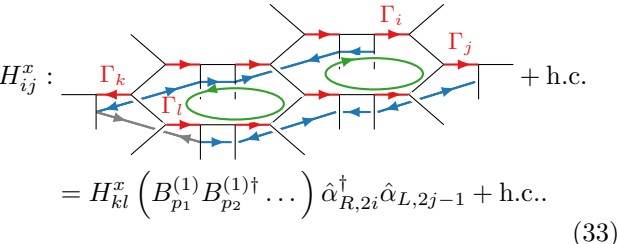

$$= H_{kl}^x \left( B_{p_1}^{(1)} B_{p_2}^{(1)\dagger} \dots \right) \hat{\alpha}_{R,2i}^\dagger \hat{\alpha}_{L,2j-1} + \text{h.c.}.$$ (33)

Here $H_{kl}^x$ is another x-link term that can be chosen to be outside of the region in which the Hamiltonian acts, so that it can be set to a constant. The product goes over all plaquette operators located between $H_{kl}^x$ and $H_{ij}^x$ and can be set to one in the ground state sector where $B_p^{(1)} = 1$ on all plaquettes. In this sector, the honeycomb Hamiltonian is quadratic in the lattice parafermions [2]:

$$H = \sum_{n=1}^{L_y} H_{1d}^{(n)} - J_x \sum_{j,n} \left( \hat{\alpha}_{R,2j}^{(n+1)\dagger} \hat{\alpha}_{L,2j-1}^{(n)} + \text{h.c.} \right),$$

$$H_{1d}^{(n)} = \sum_j \left( - J_y \omega \hat{\alpha}_{R,2j}^{(n)\dagger} \hat{\alpha}_{R,2j-1}^{(n)} \right. $$

$$\left. - J_z \omega^2 \hat{\alpha}_{R,2j}^{(n)\dagger} \hat{\alpha}_{L,2j+1}^{(n)} + \text{h.c.} \right).$$ (34)

In the above equation, $H_{1d}^{(n)}$ is the Hamiltonian of the $J_z$-$J_y$ chain that can be mapped to the Potts chain, and the $J_x$ interchain coupling is written in terms of the lattice parafermions as derived graphically in (33).

When the Potts chain is critical, the lattice parafermions contain the (anti-) holomorphic parafermion fields $\psi$, $\bar{\psi}$ of the Potts CFT, but also a non-holomorphic operator [60],

$$\hat{\alpha}_{R,j} \sim c_1 \bar{\psi} + c_2 (-1)^j \Phi_{\epsilon\bar{\sigma}},$$
$$\hat{\alpha}_{L,j} \sim d_1 \psi + d_2 (-1)^j \Phi_{\sigma\bar{\epsilon}}.$$

Here the parafermion field $\psi$ with scaling dimensions $(h,\bar{h}) = (2/3, 0)$ mixes with the operator $\Phi_{\sigma\bar{\epsilon}}$ with $(h,\bar{h}) = (1/15, 2/5)$ because they have the same conformal spin $h - \bar{h}$ modulo integers. Therefore, the $J_x$ inter-chain coupling in (34) can be expanded as

$$H^{\text{inter}} \sim \left( c_1 \bar{\psi}^\dagger + c_2 \Phi_{\epsilon\bar{\sigma}}^\dagger \right)^{(n+1)} \left( d_1 \psi - d_2 \Phi_{\sigma\bar{\epsilon}} \right)^{(n)}$$
$$\sim c_1 d_1 \bar{\psi}^\dagger \psi - c_2 d_2 \Phi_{\epsilon\bar{\sigma}}^\dagger \Phi_{\sigma\bar{\epsilon}}$$ (35)

The mixed terms $\bar{\psi}^\dagger \Phi_{\sigma\bar{\epsilon}}$ and $\Phi_{\epsilon\bar{\sigma}}^\dagger \psi$ in (35) are odd under $\mathcal{PT}$ [42] and therefore forbidden. If the interchain coupling only contained the $\bar{\psi}^\dagger \psi$ fields, the model would realize chiral parafermion topological order $\mathbb{Z}_3 \boxtimes \text{Fib}$ with gapless Potts CFT edge modes [42]. As noted in [2], the other CFT fields present in (35) can be tuned away by

adding additional interactions to the honeycomb model so that

$$\tilde{H}^{\text{inter}} = -J_x \sum_j \left( \left( \hat{\alpha}_{R,2j}^{(n+1)\dagger} + \hat{\alpha}_{R,2j-1}^{(n+1)\dagger} \right) \right.$$
$$\left. \left( \hat{\alpha}_{L,2j-1}^{(n)} + \hat{\alpha}_{L,2j-2}^{(n)} \right) + \text{h.c.} \right).$$ (36)

In the fusion surface model construction, the modified interaction term (36) can be depicted as

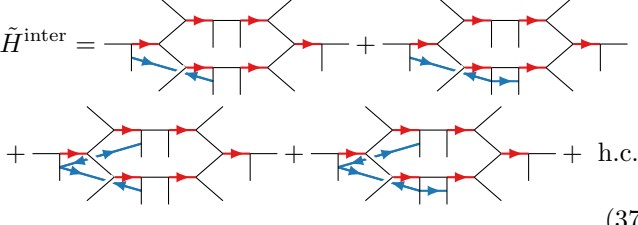

Hence, the $\mathbb{Z}_3$ fusion surface model with the modified coupling (36) instead of the $J_x$ term realizes chiral Fibonacci topological order. The interchain coupling (36) i realized in a triangular-lattice Hamiltonian also believed to exhibit chiral Fibonacci topological order [61]. In the Appendix C, we show that this model can be cast into the fusion category framework as well; it is not a fusion surface model but rather an anyon chain with long-range couplings.

For the original Hamiltonian with interchain coupling (34) and the CFT expansion (35), the coupled-wire analysis could not conclusively establish the nature of this phase, and numerics are required. Chen *et al.* [10] measure a central charge close to $c = 1$ and a topological entanglement entropy close to $\sqrt{12}$ for the model (30) at its isotropic point. Based on their results, they conclude that chiral $U(1)_{12}$ topological order is likely. However, the central charge $c = 0.8$ and topological entanglement entropy $\sqrt{3(1 + \phi^2)} \approx \sqrt{10.85}$ of chiral parafermion topological order $\mathbb{Z}_3 \boxtimes \text{Fib}$ are not too far away from the measured values.

The entanglement spectrum proves a useful tool for distinguishing different types of topological order. Namely, the low-lying entanglement energies of a ground state with chiral topological order are characterized by the CFT of its gapless edge modes [62, 63]. It was used by Stoudenmire *et al.* [61] to distinguish between a chiral parafermion and a chiral $U(1)_6$ phase in their $\mathbb{Z}_3$ model. They found signatures of chiral $U(1)_6$ topological order in the entanglement spectrum of their model in the square lattice limit. Their Hamiltonian in this limit has almost the same parafermion description (33) as the honeycomb model (30) in fixed $B_p = 1$ sectors (the only difference being that their inter-chain coupling is invariant under translations by one and not by two sites).

To gain more insight into the isotropic phase of our $\mathbb{Z}_3$ model, we measure the entanglement spectrum of the ground state on an infinite cylinder, using the DMRG package Tenpy [64]. More details on the numerical simulations are collected in Appendix D. The partition function of the Potts CFT with free boundary conditions is

given by

$$Z_{\text{f,f}}^{\text{Potts}}(q) = q^{-c/24}\big(1 + 2q^{2/3} + 2q^{5/3} + q^2$$
$$+ 4q^{8/3} + 2q^3 + \dots\big). \tag{38}$$

The degeneracies of the lowest entanglement energies in a chiral parafermion phase are expected to match the coefficients $(1, 2, 2, 1, (2, 2) \dots)$ in the partition function. The $(2, 2)$ notation indicates that the four-fold degeneracy can be split into two two-fold degeneracies by finite size effects, as the corresponding term $4q^{8/3}$ in the partition function is the sum of contributions $2q^{2^2 \cdot 2/3}$ from the $\chi_{2/3}$ character and $2q^{5/3+1}$ from the $\chi_{5/3}$ character. This pattern is indeed what we observe in Fig. 5 for the ratios of degeneracies on cylinders of circumferences $L_y = 2, 3, 4$. The absolute degeneracies in the $L_y = 3, 4$ plots are higher due to the conserved plaquette operators crossing an entanglement cut, as observed in [65] for the original Kitaev honeycomb model. The $L_y = 2, 4$ entanglement spectra are computed across a different bond of the matrix product state than the $L_y = 3$ spectrum, due to an even-odd effect on the cylinder, see Appendix D for details. We also checked that these degeneracies remain the same for various $J_z < 1$. For a chiral $U(1)_{12}$ phase, we would expect a different degeneracy pattern $(1, 2, 1, 2, 2, \dots)$.

Considering the clear signatures of chiral topological order observed in Chen *et al.* [10] as well as the parafermion CFT degeneracies in the entanglement spectra presented in Fig. 5, it seems likely that the phase realizes chiral parafermion topological order. One does not need to add a magnetic field or an analog of $V^{(3)}$ from (22) to obtain chiral topological order; the time-reversal symmetry breaking arising from the braiding in the fusion surface models appears sufficient.

## VI. THE FIBONACCI FUSION SURFACE MODEL

The Tambara-Yamagami categories give rise to very special fusion surface models with dynamical degrees of freedom located only on half of the planar edges, as discussed in Sections IV and V. To explore more generic fusion surface models, where degrees of freedom live on all planar edges of the honeycomb fusion tree, we here investigate the model built from the Fibonacci fusion category. This novel 2+1d Fibonacci model preserves a non-invertible 1-form symmetry and explicitly breaks time-reversal symmetry. Through a coupled-wire analysis, we show that with appropriately fine-tuned interactions, the model likely exhibits chiral topological order with tricritical Ising edge modes.

(i) $L_y = 2$, $D = 200$  (ii) $L_y = 3$, $D = 400$

(iii) $L_y = 4$, $D = 800$

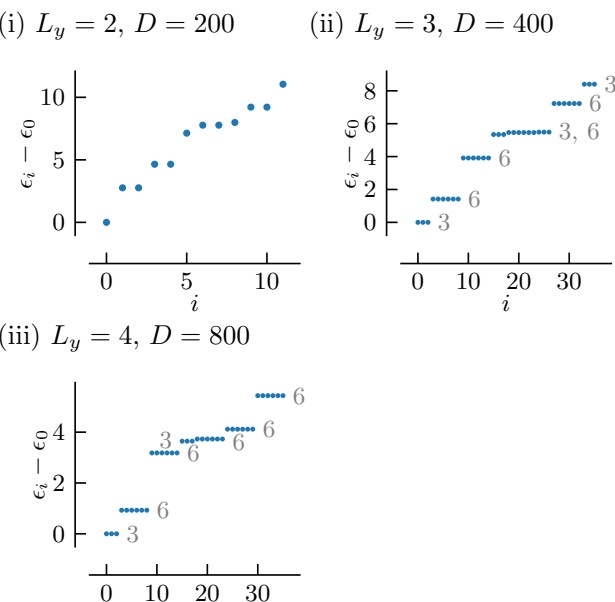

FIG. 5. Entanglement energies $\epsilon_i$ of the $\mathbb{Z}_3$ honeycomb model (30) with $J_x = J_y = J_z = 1$ on an infinite cylinder for different circumferences $L_y$ and bond dimensions $D$; the degeneracies are written in gray. For chiral Fibonacci topological order, a degeneracy pattern of $(1, 2, 2, 1, (2, 2), 1, \dots)$ is expected [61].

## A. Constrained Hilbert space, broken time-reversal and non-invertible 1-form symmetry

The Fibonacci category contains two self-dual objects $\{1, \tau\}$ with fusion rule $\tau \otimes \tau = 1 \oplus \tau$. In our Fibonacci fusion surface model, all vertical legs of the fusion tree are labeled by the object $\tau$. The Hilbert space is spanned by the states $|\{\Gamma_i, \Gamma_{ijk}\}\rangle$ with degrees of freedom $\Gamma_i, \Gamma_{ijk} \in \{1, \tau\}$ on all planar edges of the honeycomb fusion tree.

$$|\{\Gamma_i, \Gamma_{ijk}\}\rangle = $$

At each trivalent vertex, the Fibonacci fusion rule must be obeyed, resulting in a constrained Hilbert space. The Fibonacci fusion surface Hamiltonian $H = \sum_p H_p$ has the same structure (1) as Kitaev's honeycomb model, with $H_p$ acting as

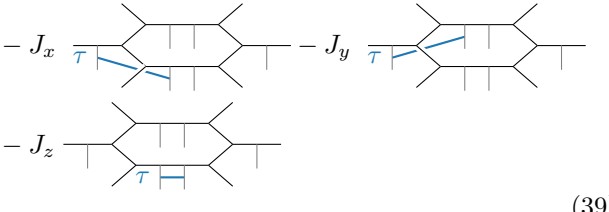

$$\tag{39}$$

The x-link and y-link terms now couple seven degrees of freedom. Because $\tau$ has a trivial Frobenius-Schur indicator, the Hamiltonian is automatically hermitian [1]. In

Appendix E, the fusion diagrams in (39) are evaluated explicitly to compute the Hamiltonian in operator form.

The Fibonacci Hamiltonian breaks time-reversal symmetry explicitly. Although the z-link term is real, the x-link term includes complex operators such as

$$R_\tau^{\tau\tau}\sigma_{ijk}^+ + (R_\tau^{\tau\tau})^*\sigma_{ijk}^- = \begin{pmatrix} 0 & (R_\tau^{\tau\tau})^* \\ R_\tau^{\tau\tau} & 0 \end{pmatrix}_{ijk},$$

where $R_\tau^{\tau\tau} = e^{3\pi i/5}$. If a unitary $U$ existed such that $UH_pU^\dagger = H_p^*$, it would need to map $\sigma_{ijk}^\pm \to \sigma_{ijk}^\mp$ in the x-link term, while also commuting with the $n_{ijk} = \text{diag}(0,1)_{ijk}$ matrix in the real z-link term. Such a transformation would change the commutation relations between $n$ and $\sigma^\pm$, and so cannot be implemented by a unitary operator. An anti-unitary time-reversal symmetry $U\mathcal{K}$ is therefore ruled out.

By construction, the Fibonacci fusion surface model (39) has a 1-form symmetry generated by fusing a $\tau$-line to the lattice from above, as well as conserved plaquette operators $B_p^{(\tau)}$. These symmetries are non-invertible because they obey the same fusion algebra as the object $\tau$ in the input category, for instance $\left(B_p^{(\tau)}\right)^2 = \mathbb{I} + B_p^{(\tau)}$.

### B. Doubled Fibonacci topological order and weakly coupled tricritical Ising chains

To understand the Fibonacci model (39) in the anisotropic limit $J_z \gg J_x, J_y$, we apply the results derived in Section III B. When $J_z \to \infty$, it is known from the completely staggered ferromagnetic Fibonacci chain that there are two ground states, $|1\tau 1\rangle$ and $|\tau\tilde\tau\tau\rangle$, on each z-link [17], where $|\tilde\tau\rangle = \phi^{-1}|1\rangle + \phi^{-1/2}|\tau\rangle$ and $\phi$ denotes the golden ratio. Each z-link can thus be replaced by a single horizontal edge labeled by 1 or $\tau$, cf. (9):

$$
\begin{array}{c}
\text{(40 diagram)}
\end{array}
\tag{40}
$$

This substitution results in a highly degenerate ground state subspace that forms a Fibonacci string-net on the honeycomb lattice.

For sufficiently large systems, the lowest order Hamiltonian generated by perturbation theory in $J_x, J_y \ll J_z$ is the Levin-Wen plaquette operator,

$$H^{\text{eff}} \sim \frac{J_x^2 J_y^2}{J_z^3} \quad \boxed{\tau} \tag{41}$$

This Fibonacci Levin-Wen Hamiltonian, along with a magnetic field perturbation, has been studied in [66–68]. It realizes doubled Fibonacci topological order. The same holds for the limit $J_x \to \infty$, since the energy spectrum

is invariant under exchanging $J_z$ and $J_z$ (for a symmetric geometry $L_x = L_y$ and suitable toroidal boundary conditions, as we checked numerically).

In the decoupled limit $J_x = J_y$ and $J_z = 0$, the model (39) reduces to $L_y$ $J_x$-$J_y$ chains with Hamiltonian

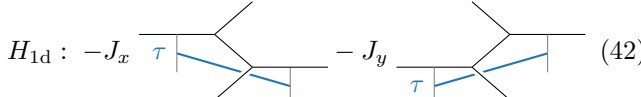

$$H_{1\text{d}}: \quad -J_x \quad \cdots \quad - J_y \quad \cdots \tag{42}$$

As explained in Sec. III C, the $J_x$-$J_y$ chain (42) is unitarily related to the Fibonacci anyon chain with a sum over boundary conditions. Large degeneracies arise because of the plaquette 1-form symmetries, as illustrated in App. G for the Ising fusion surface model. With uniform couplings this chain is the Hamiltonian limit [69] of a critical point of the integrable model of hard squares with diagonal interactions [70, 71], which is the $A_4$ case of the Andrews-Baxter-Forrester RSOS models [20]. It was reformulated in terms of fusion-category data and (re)named the "golden chain" [15].

Writing this Fibonacci chain in terms of a fusion category yields a remarkable insight: the chain is invariant under a "topological symmetry", generated by fusing a $\tau$-line to the fusion tree from above [15]. This symmetry provides a non-trivial generalisation of Kramers-Wannier duality, and now is recognized as a canonical example of a non-invertible, categorical or generalized symmetry [17, 18, 72–74], meaning that it cannot be represented by a unitary matrix. Instead, non-invertible symmetries and dualities are often conveniently implemented by matrix product operators. (Somewhat ironically, this symmetry generator does have an inverse, but we continue to use the current parlance.)

With uniform ferromagnetic couplings, the Fibonacci anyon chain is critical and described by the tricritical Ising conformal field theory [69–71, 75]. The same must hold for the $J_x$-$J_y$ chain (42) with $J_x = J_y > 0$, as boundary conditions do not change the gap of the system. The tricritical Ising CFT contains a topological defect line corresponding to the non-invertible symmetry on the lattice, which obeys the same fusion algebra. Of the six chiral primary fields, only $\sigma'$ with scaling dimension $h = 7/16$ and $\epsilon''$ with scaling dimension $h = 3/2$ are commuting with the topological defect line [15].

The $J_z$ term, which couples adjacent chains, commutes with the non-invertible 1-form symmetry of the fusion surface model. In the continuum limit, it can be expanded in terms of CFT fields of the critical chains it couples. Due to the non-invertible 1-form symmetry, this expression can only include $\sigma'$ and $\epsilon''$. The most relevant fields in the expansion have zero conformal spin and are given by:

$$H^z \sim a_1 \Phi_{\sigma'\bar\sigma'}^{(n,n+1)} + a_2 \Phi_{\sigma'\sigma'}^{(n,n+1)} + a_3 \Phi_{\sigma'\bar\sigma'}^{(n)}\Phi_{\sigma'\bar\sigma'}^{(n+1)} \tag{43}$$

The notation $\Phi_{\sigma'\bar\sigma'}^{(n,n+1)}$ (instead of $\sigma_L'^{(n)}\sigma_R'^{(n+1)}$) reflects that the chiral fields $\sigma'$ cannot be realized separately as

local or semi-local lattice operators [60]. The fields with coefficients $a_1$ and $a_2$ in (43) combine the left and right $\sigma'$ fields from two adjacent chains, whereas the $a_3$ term is a product of the non-chiral $\Phi_{\sigma'\bar{\sigma}'}$ fields from both chains. Since time-reversal symmetry is explicitly broken in the lattice Hamiltonian, we must have $a_1 \neq a_2$ in the expansion. Due to the presence of multiple CFT fields in (43), the nature of the resulting phase remains unclear. It could be gapless or (chiral) topological order Fib $\boxtimes \mathcal{C}$, constrained by the anomalous non-invertible 1-form symmetry.

If only the first term $\Phi_{\sigma'\bar{\sigma}'}^{(n,n+1)}$ appeared in (43), the model would exhibit chiral topological order with tricritical Ising edge modes, by a similar argument as presented in [42]: The tricritical Ising CFT perturbed by $\Phi_{\sigma'\bar{\sigma}'}$ is gapped with two degenerate (but not symmetry-related) ground states [76], implying that the bulk of the coupled chain system is gapped. Since the coupling $\Phi_{\sigma'\bar{\sigma}'}^{(n,n+1)}$ does not contain the right-moving CFT fields of the bottom chain and the left-moving CFT fields of the top chain, gapless tricritical Ising edge modes remain. It is plausible that the $a_2$ and $a_3$ terms in (43) could be tuned away by adding different lattice couplings between adjacent chains, such as those depicted in (37) in the context of the $\mathbb{Z}_3$ model. The coupled-wire system with the $\Phi_{\sigma'\bar{\sigma}'}^{(n,n+1)}$ coupling, which cannot be easily decomposed into a product of two lattice operators from individual chains, appears to be unexplored. Li *et al.* [77] studied a system of coupled GSV chains [78] with tricritical Ising edge modes, but their coupling does not seem to respect the non-invertible Fibonacci symmetry, yielding a different coupled-wire field theory.

To support our phase diagram analysis, we compute the lowest energy levels in small Fibonacci models with $L_y = 2$, $L_x = 2, 3$. This was done with the exact diagonalization package Quspin [79]. We choose periodic boundary conditions in both directions, as depicted in (45). The energy gaps for $J_x = J_y = 1$ and $J_z \in [0,3]$ are shown in Fig. 6. The plot indicates two distinct regimes separated by a phase transition, consistent with our qualitative understanding of an anisotropic phase and a weakly coupled chains phase. The ground state is always two-fold degenerate, suggesting that the 1-form symmetry around one of two incontractible cycles around the torus is spontaneously broken. A closer examination of the ground state energies in the large $J_z$ limit, shown in Fig. 7, reveals that the leading contributions are

$$E_{\text{GS}} = -J_z L_x L_y E_{\text{GS}}^{(\text{z-link})} - \frac{C}{J_z} + \mathcal{O}\left(\frac{1}{J_z^2}\right) \quad (44)$$

The $1/J_z$ contribution arises from incontractible loops around the torus, generated at second order in perturbation theory when $L_y = 2$ or $L_x = 2$. Schematically,

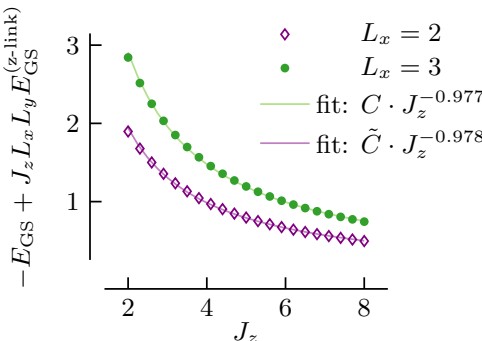

$$H^{(\text{eff})} \sim \frac{J_x J_y}{J_z} \quad , \quad (45)$$

with the dotted edges wrapping around the torus. Due to these $1/J_z$ terms, which dominate over the $1/J_z^3$ Levin-Wen projector terms, we are unable to observe four degenerate ground states characteristic of doubled Fibonacci topological order.

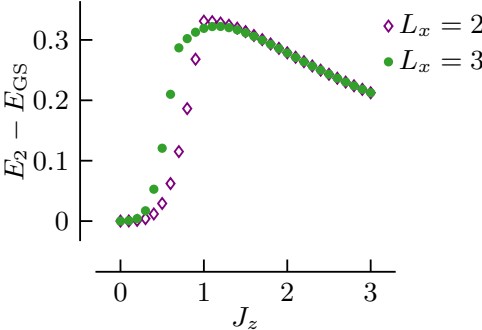

FIG. 6. Energy gap of the $L_y = 2$ Fibonacci models on a torus for $J_x = J_y = 1$

FIG. 7. Ground state energies of the Fibonacci model on a $L_y = 2$ torus for $J_x = J_y = 1$, with the zeroth-order contribution $-J_z L_x L_y E_{\text{GS}}^{(\text{z-link})}$ subtracted and the second-order contribution fitted, cf. (44).

If the weakly coupled chains phase is described by chiral topological order, its ground state must break time-reversal symmetry. To test this hypothesis, we define a time-reversal invariant interpolation of the Fibonacci model,

$$H_p = H^z + \lambda(H^x + H^y) + (1-\lambda)(H^x + H^y)^* \quad (46)$$

A similar interpolation was discussed in [56] for the $\mathbb{Z}_3$ generalization of Kitaev's honeycomb model. The ground state energies of the interpolated Fibonacci Hamiltonian (46) are shown in Fig. 8. It is conceivable that a phase transition occurs at $\lambda = 0.5$, though the plot is not conclusive and larger system sizes would be necessary to confirm the existence of a transition. A phase transition at $\lambda = 0.5$ would be evidence for chiral topological order, as it implies that the ground states of the $\lambda = 0$ and $\lambda = 1$ Hamiltonians cannot be adiabatically connected.

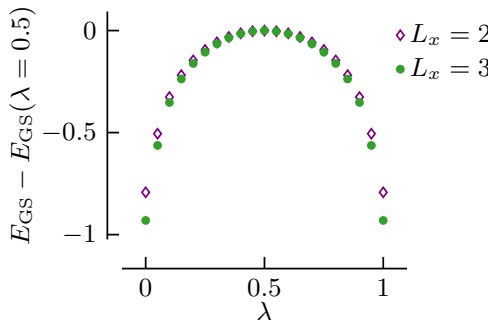

FIG. 8. Ground state energies of the interpolated Hamiltonian (46) on a $L_y = 2$ torus with $J_x = J_y = \lambda$, $J_z = 0.6$

## VII. CONCLUSIONS

We have shown how interesting and important 2d quantum lattice models for chiral topological order can be treated in a common framework. The models are defined using data from braided fusion categories, giving a natural generalization of the construction of 2d classical and their quantum anyon-chain limits. Defining the models in this fashion builds in a useful symmetry structure, including conserved local operators, as well as non-invertible 1-form symmetries. It also allows us to introduce new models that have all the ingredients for chiral topological order.

In particular, we used the fusion surface model construction of Inamura and Ohmori [1] to systematically build generalizations of Kitaev's honeycomb model from braided fusion categories. By construction they have mutually commuting conserved plaquette operators and 1-form symmetries generalizing those of Kitaev's [3]. When the braiding phases of the input category are nontrivial, the 1-form symmetries are anomalous, requiring them to be spontaneously broken or the phase to be gapless. In addition, their phase diagrams share common features with Kitaev's honeycomb model. In the anisotropic limit, the fusion surface models reduce to Levin-Wen stringnets. In the isotropic limit, they are described by weakly coupled anyon chains and potentially exhibit chiral topological order.

A certain time-reversal breaking perturbation of Kitaev's honeycomb model results in chiral Ising topological order. We showed that this perturbation can be incorporated into the categorical framework, and so provide a positive answer to the question raised by Inamura and Ohmori [1] about the realization of chiral topological order in fusion surface models. Moreover, we explained how closed $\sigma$-loops fused to the lattice from above implement non-invertible 1-form dualities, as they introduce lattice dislocations. At the endpoints of open $\sigma$-strings, non-abelian twist defects arise.

We then showed how the $\mathbb{Z}_N$-invariant fusion surface model built from the $\mathbb{Z}_N$ Tambara-Yamagami category is closely related to the $\mathbb{Z}_N$-generalization of Kitaev's hon-

eycomb model introduced in Barkeshli *et al.* [2]. We provide numerical evidence indicating the presence of chiral parafermion topological order near the isotropic point. The coupled-wire analysis also points to chiral parafermion topological order, but only when one of the two relevant CFT fields is tuned away by adding additional interactions. Hence, the numerical results suggest that the coupled-wire system may be less sensitive to the presence of additional relevant CFT fields than previously thought, warranting further investigation.

The Fibonacci category gives rise to a novel honeycomb model with a non-invertible 1-form symmetry and explicitly broken time-reversal symmetry. In the anisotropic limit, this model supports double Fibonacci topological order. The isotropic phase, qualitatively described by weakly coupled critical Fibonacci chains, remains to be conclusively understood. Our numerical work at minimum shows that chiral topological order is possible. Likely, large-scale numerical simulations such as infinite DMRG or PEPS on the constrained Hilbert space are required to establish it convincingly.

When the eigenvalues of the plaquette operators are fixed, Kitaev's honeycomb model is exactly solvable via a mapping to free fermions. While the $\mathbb{Z}_3$ and Fibonacci generalizations explored do by construction possess a great deal of symmetry, they do not appear to be integrable. Despite this, it remains an intriguing open question whether there exist similar fusion surface models that exhibit some form of higher-dimensional integrability, whether that be through free fermions or some other mechanism.

### Acknowledgments

We thank Thomas Wasserman for very helpful explanations of $G$-crossed braiding and Fiona Burnell, David Penneys, and Sakura Schafer-Nameki for interesting discussions. This work has been supported in part by the EPSRC Grant no. EP/S020527/1.

## Appendix A: Derivation of the Ising fusion surface model

In the Ising category, the quantum dimensions of the objects are $d_1 = d_0 = 1$ and $d_\sigma = \sqrt{2}$ and the non-trivial F-symbols and R-symbols are

$$[F_1^{\sigma 1\sigma}]_{\sigma\sigma} = [F_\sigma^{1\sigma 1}]_{\sigma\sigma} = -1, \ F_\sigma^{\sigma\sigma\sigma} = \frac{1}{\sqrt{2}}\begin{pmatrix} 1 & 1 \\ 1 & -1 \end{pmatrix}, \ R_0^{11} = -1, \ R_\sigma^{1\sigma} = R_\sigma^{\sigma 1} = -i, \ R_0^{\sigma\sigma} = e^{\frac{-i\pi}{8}}, \ R_1^{\sigma\sigma} = e^{\frac{3\pi i}{8}}.$$

The z-link term in the Hamiltonian (19) can be evaluated by fusing the 1-line to the horizontal edge, using F-moves, and removing bubbles:

In operator form, this is

$$H^z_{klm,mpq} = Z_{klm} Z_{mpq}.$$

The x-link term can be evaluated similarly,

Here $[\cdot]_2$ denotes addition modulo 2, and only the non-trivial F-symbols are written down. As an operator,

$$H^x_{ijk,klm} = i Z_{klm} X_{klm} X_{ijk} = -Y_{klm} X_{ijk}.$$

Analogous to the x-link term, the y-link term yields

In operator form,

$$H^y_{ijk,jno} = (-i) Z_{ijk} X_{ijk} X_{jno} = Y_{ijk} X_{jno}.$$

The entire Hamiltonian is then

$$H = -J_x \sum_{b,a \in \text{x-link}} (-Y_b X_a) - J_y \sum_{a,c \in \text{y-link}} Y_a X_c - J_z \sum_{b,d \in \text{z-link}} Z_b Z_d,$$

with the vertices of the honeycomb lattice now labeled by single letters $a, b, \dots$ for brevity.

## Appendix B: Derivation of the $\mathbb{Z}_N$ Tambara-Yamagami fusion surface model

The F-symbols and R-symbols of the Tambara-Yamagami category $\mathcal{C}_G = \mathcal{C}_0 \oplus \mathcal{C}_1$ with $\mathcal{C}_0 = \mathbb{Z}_N^{(r)}$, $\mathcal{C}_1 = \{\sigma\}$, $N$ odd and $0 < r \leq N - 1$ are discussed in Section V A. The action of the z-link term on the qudits $\Gamma_{klm} \in \{0, 1, \dots, N-1\}$ is depicted below,

In operator form,

$$H^z_{klm,mpq} = Z^{r\dagger}_{klm} Z^r_{mpq}.$$

The Hamiltonian acts on the x-links as:

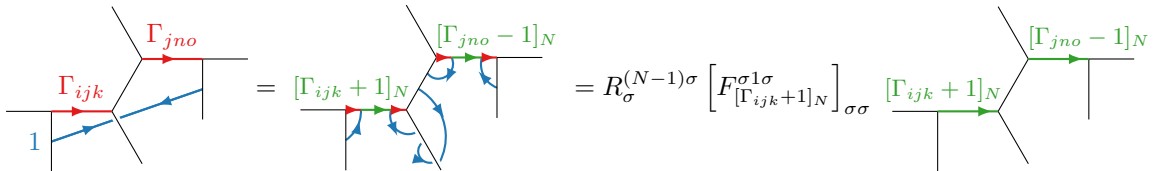

As an operator, the x-link term is equal to

$$H^x_{ijk,klm} = (-1)^{rN} e^{-\frac{i\pi r}{N}} X_{ijk} Z^r_{klm} X^\dagger_{klm}$$

The action of the Hamiltonian on the y-links is given by:

$$\Gamma_{jno} \quad = \quad [\Gamma_{jno}-1]_N \quad = R_\sigma^{(N-1)\sigma} \left[ F^{\sigma 1\sigma}_{[\Gamma_{ijk}+1]_N} \right]_{\sigma\sigma} \quad [\Gamma_{jno}-1]_N$$

As an operator,

$$H^y_{ijk,jno} = (-1)^{rN} e^{\frac{i\pi p}{N}} Z^r_{ijk} X_{ijk} X^\dagger_{jno}.$$

The entire $\mathbb{Z}_N$ fusion surface model Hamiltonian is then

$$H = -J_x \sum_{b,a\in\text{x-link}} (-1)^{rN} e^{-\frac{i\pi r}{N}} X_a Z^r_b X^\dagger_b - J_y \sum_{a,c\in\text{y-link}} (-1)^{rN} e^{\frac{i\pi r}{N}} Z^r_a X_a X^\dagger_c - J_z \sum_{b,d\in\text{z-link}} Z^{r\dagger}_b Z^r_d + \text{ h.c.} \tag{B1}$$

Next we discuss the connection of the fusion surface model (B1) to the $\mathbb{Z}_N$ generalization of Kitaev's honeycomb model proposed in [2]. After a unitary transformation that sends $X^\dagger_b \to (-1)^{rN} e^{\frac{i\pi r}{N}} Z^{r\dagger}_b X^\dagger_b$ for all qudits on one sublattice, (B1) becomes

$$H = -J_x \sum_{b,a\in\text{x-link}} X_a X^\dagger_b - J_y \sum_{a,c\in\text{y-link}} \omega^r Z^r_a X_a Z^{r\dagger}_c X^\dagger_c - J_z \sum_{b,d\in\text{z-link}} Z^{r\dagger}_b Z^r_d + \text{ h.c.}$$

Then applying unitary charge conjugation (28) to the same sublattice yields

$$H = -J_x \sum_{b,a\in\text{x-link}} X_a X_b - J_y \sum_{a,c\in\text{y-link}} \omega^r Z^r_a X_a Z^r_c X_c - J_z \sum_{b,d\in\text{z-link}} Z^r_b Z^r_d + \text{ h.c.}$$

For $r=1$ and $r=N-1$, this is the Hamiltonian studied in [2], except for the additional complex phase in the y-link term.

## Appendix C: Square lattice $\mathbb{Z}_3$ model in the fusion category framework

The $\mathbb{Z}_3$ symmetric lattice model studied in [61] is believed to realize chiral parafermion topological order $\mathbb{Z}_3 \boxtimes$ Fib in its triangular lattice limit. In terms of lattice parafermions, it can be expressed as

$$H = \sum_{n=1}^{L_y} H^{(n)}_{1d} + \sum_n H^{(n,n+1)}_{\text{inter}}, \quad \text{with } H^{(n)}_{1d} = -t_3 \sum_j \left( \omega \hat{\alpha}^{(n)\dagger}_{R,j+1} \hat{\alpha}^{(n)}_{R,j} + \text{h.c.} \right) \text{ and}$$

$$H^{(n,n+1)}_{\text{inter}} = -\sum_j \left( t_1 \omega \hat{\alpha}^{(n)\dagger}_{L,j} \hat{\alpha}^{(n+1)}_{R,j} + t_2 \omega \hat{\alpha}^{(n)\dagger}_{L,j} \hat{\alpha}^{(n+1)}_{R,j-1} + \text{h.c.} \right). \tag{C1}$$

The 1d Hamiltonians are decoupled Potts models, and their lattice parafermions are known in the fusion category framework, cf. (32). The complex phases in (C1) are necessary to have charge conjugation symmetry. The triangular lattice limit corresponds to choosing $t_1 = t_2$ in (C1), so that the field theory expansion of the inter-chain coupling only contains the parafermion operator $\bar{\psi}^\dagger \psi$, cf. (35). Writing down the Hamiltonian (C1) in operator form requires choosing a parafermion path. We take the same path that was used for the numerical simulations in [61], depicted in Fig. 20 in [61], and also fix $L_y = 4$ and open boundary conditions in the y-direction. With the parafermion definitions in [61], one unit cell of the Hamiltonian (C1) can then be written as

$$
\begin{aligned}
H = {} & t_1 \left( \tau_1 + \sigma_2 \sigma_1^\dagger + \tau_2 \right) + t_2 \left( \sigma_1 \tau_2 \sigma_3^\dagger + \sigma_2 \tau_2 \tau_3^\dagger \sigma_3^\dagger + \sigma_2 \tau_3^\dagger \sigma_4^\dagger \right) \\
& + t_3 \left( \omega \sigma_1 \tau_1 \tau_2 \sigma_3^\dagger + \omega^2 \sigma_1 \tau_2 \tau_3^\dagger \sigma_3^\dagger + \omega \sigma_2 \tau_2 \tau_3^\dagger \sigma_4^\dagger + \omega^2 \sigma_2 \tau_3^\dagger \tau_4^\dagger \sigma_4^\dagger \right) + \text{h.c.}
\end{aligned}
\tag{C2}
$$

Using the graphical expressions (32) for the parafermions, the unit cell Hamiltonian (C1) can be depicted as follows:

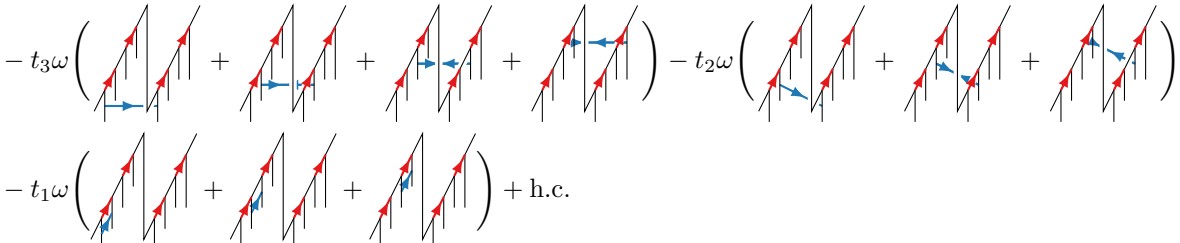

So the Hamiltonian (C1) can be regarded as an anyon chain with long range interactions. Unlike the fusion surface models, such anyon chain models do not have conserved plaquette operators. An anyon chain model very similar to (C1) but with coupling

$$
-t_4 \omega \sum_j \left( \hat{\alpha}_{L,2j-1}^{(n)} \hat{\alpha}_{R,2j}^{(n+1)\dagger} + \text{h.c.} \right) = -t_4 \omega \sum_j \left( \vcenter{\hbox{}} + \vcenter{\hbox{}} \right)_{2j} + \text{h.c.}
$$

corresponding to the parafermion coupling (34) has the same energies as the $\mathbb{Z}_3$ fusion surface model (26) (but smaller degeneracies).

## Appendix D: Details on the DMRG simulations of the $\mathbb{Z}_3$ models

We use Tenpy [64] for infinite DMRG simulations on the cylinder. Before computing the entanglement spectra of the $\mathbb{Z}_3$ honeycomb model (30), we reproduce the entanglement spectra of the $\mathbb{Z}_3$ square lattice model (C2) studied in [61] to ensure that our numerical methods are reliable. For the square lattice model (C2), the DMRG unit cell is $L_x = 2$ and $\mathbb{Z}_3$ symmetry is conserved in the simulation (note that the conservation of energies can influence which entanglement energies appear in the spectrum [80]). The results are shown in Fig. 10 for (i) $t_3 = 1$ and (ii) $t_3 = -1$, both with different $t_1 = t_2$. While the values of the entanglement energies vary with $t_1 = t_2$, their degeneracies remain the same. When $t_3 = 1$, the degeneracy pattern $(1, 2, 2, 1, \dots)$ is consistent with chiral parafermion order, see (38). When $t_3 = -1$, the pattern $(1, 2, 1, 4, \dots)$ is consistent with chiral $U(1)_6$ topological order. The presence of chiral $U(1)_6$ topological order is not surprising because the square lattice model with negative $t_3$ reduces to decoupled antiferromagnetic Potts models described by the $U(1)_6$ CFT when $t_1 = t_2 = 0$. In the chiral $U(1)_6$ phase, there are two ground states with different entanglement energies [61], so randomizing the initial state is essential to find the ground state that gives rise to the $(1, 2, 1, 4, \dots)$ pattern.

Next we simulate the the $\mathbb{Z}_3$ honeycomb model and choose a unit cell of $L_x = 3$ following [10]. First we checked that the fusion surface models (29) with $N = 3$, $p = 1$ and $p = 2$ have the same energies and entanglement spectrum as the model (30) on an infinite cylinder (albeit having slightly different finite-size energies). Therefore we focus on the Hamiltonian (30) in subsequent simulations. Its entanglement spectrum at the isotropic point is shown in Fig. 5, and it shows the $(1, 2, 2, 1, \dots)$ degeneracies characteristic of chiral parafermion topological order. On the $L_y = 2$ and $L_y = 4$ cylinder, the entanglement cut is across bond 0, which is the canonical choice, but on the $L_y = 3$ cylinder, the cut goes across bond 1 in order to recover the same degeneracies, see Fig. 9. This is probably due to an even vs. odd effect of the honeycomb model on an infinite cylinder. The entanglement spectra for $L_y = 3$, $J_y = J_z = 1$ and different $J_x$ are shown in Fig. 11(i). Also, we show the entanglement spectra for $L_y = 4$ and $J_y = J_z = -1$ in

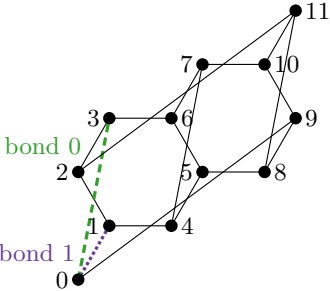

FIG. 9. The geometry of the MPS used in our Tenpy infinite DMRG simulations for $L_x = 3$ and $L_y = 2$. The entanglement spectra are computed across bond 0 (dashed green line) and across bond 1 (dotted violet line) for even and odd $L_y$ respectively.

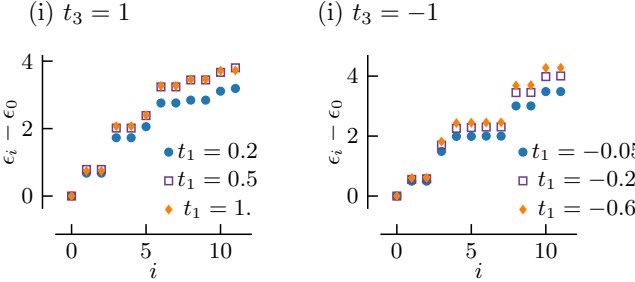

FIG. 10. Entanglement energies $\epsilon_i$ of the $\mathbb{Z}_3$ square lattice model (C2) with (i) $t_3 = 1$ (chiral parafermion phase) and (ii) $t_3 = -1$ (chiral $U(1)_6$ phase) and different $t_1 = t_2$ on an infinite cylinder with bond dimension $D = 400$.

Fig. 11(ii) and observe that the degeneracies follow the $(1, 2, 1, 4, \dots)$ pattern expected for $U(1)_6$ (though $U(1)_{12}$ has very similar degeneracies $(1, 2, 1, 2, 2, \dots)$ and cannot be ruled out).

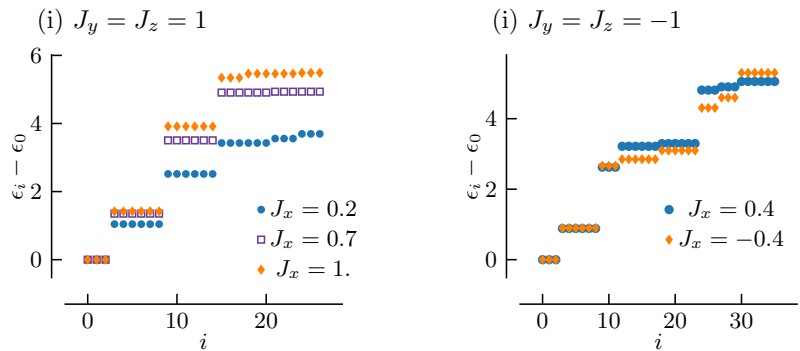

FIG. 11. Entanglement energies $\epsilon_i$ of the $\mathbb{Z}_3$ honeycomb model (30) with (i) $J_y = J_z = 1$ (likely chiral parafermion phase), $D = 800$, $L_y = 3$ and the entanglement cut across bond 1, (ii) $J_y = J_z = -1$ (likely chiral $U(1)_6$), $D = 1000$, $L_y = 4$, entanglement cut across bond 0.

## Appendix E: Derivation of the Fibonacci fusion surface model

To derive the Hamiltonian explicitly, we use the F-symbols and R-symbols of the Fibonacci category,

$$F_\tau^{\tau\tau\tau} = \begin{pmatrix} \phi^{-1} & \phi^{-1/2} \\ \phi^{-1/2} & -\phi^{-1} \end{pmatrix} = (F_\tau^{\tau\tau\tau})^{-1}, \; R_1^{\tau\tau} = e^{-4\pi i/5}, \; R_\tau^{\tau\tau} = e^{3\pi i/5}, \; d_\tau = \phi,$$

where $\phi = (1+\sqrt{5})/2$ is the golden ratio. In addition, we define the following matrix operators in the $\{1,\tau\}$ basis,

$$\sigma^x = \begin{pmatrix} 0 & 1 \\ 1 & 0 \end{pmatrix}, \ n = \begin{pmatrix} 0 & 0 \\ 0 & 1 \end{pmatrix}, \ \tilde{n} = \begin{pmatrix} 1 & 0 \\ 0 & 0 \end{pmatrix}, \ \sigma^- = \begin{pmatrix} 0 & 1 \\ 0 & 0 \end{pmatrix}, \ \sigma^+ = \begin{pmatrix} 0 & 0 \\ 1 & 0 \end{pmatrix}.$$

The z-link term of the Fibonacci fusion surface model (39) can be depicted as

$$\frac{\begin{matrix} a & b & c \end{matrix}}{\tau} = \sum_{b'} \sqrt{\frac{d_{b'}}{d_b d_\tau}} \quad \overset{b'}{\phantom{x}} = \sum_{b'} [F_a^{b'\tau\tau}]_{b\tau} [F_c^{\tau\tau b}]_{\tau b'} \sqrt{d_\tau} \tag{E1}$$

Here we denote the degrees of freedom by single arabic letters $a$, $b$, ... instead of $\Gamma_i$, $\Gamma_{ijk}$ as previously to avoid cluttering notation. Evaluating the above fusion diagram using the F-symbls and R-symbols of the Fibonacci category shows that the z-link term acting on the constrained Hilbert space is equal to

$$H^z = \phi^{1/2} \tilde{n}_a n_b \tilde{n}_c - \phi^{-1/2} (\tilde{n}_a n_b n_c + n_a n_b \tilde{n}_c) + n_a n_c (\sigma_b^x + \phi^{-3/2} n_b). \tag{E2}$$

Up to an additive and a multiplicative constant, the z-link Hamiltonian (E2) is the same as the golden chain Hamiltonian [15]. The x-link Hamiltonian can be depicted as

$$\overset{\begin{matrix} g & f & e \end{matrix}}{\underset{\begin{matrix} c \end{matrix}}{\tau \ \ d \ \ b \ \ a}} = \sum_{b',c',d',f'} \sqrt{\frac{d_{b'} d_{c'} d_{d'} d_{f'}}{d_x^4 d_b d_c d_d d_f}} \quad \overset{f'}{\phantom{x}} \tag{E3}$$

$$= \sum_{b',c',d',f'} (R_{c'}^{c\tau})^* [F_a^{\tau x b}]_{\tau b'} [F_d^{b' x c}]_{bc'} [F_{b'}^{c x d}]_{c'd'} [F_e^{d' x f}]_{df'} [F_g^{f' x \tau}]_{f\tau} \sqrt{d_\tau} \quad \overset{\begin{matrix} g & f' & e \end{matrix}}{\underset{c}{\phantom{x} d' \ b' \ a}}$$

Here the factors from bubble removal essentially cancel the factors from fusing the $\tau$-line to the lattice. The Hamiltonian is diagonal in $a$, $c$, $e$ and $g$, and so decomposes into separate blocks for fixed $a, c, e, g$. We will use the identities

$$R_1^{\tau\tau} + R_\tau^{\tau\tau} \phi^{-1} = -1, \ R_\tau^{\tau\tau} + R_1^{\tau\tau} \phi^{-1} = R_1^*, \ (R_1 - R_\tau)\phi^{-1} = R_\tau^*.$$

When we fix $c = e = 1$ in (E3), it is enforced that $b = d = f$ and so we recover the z-link Hamiltonian (E2).

$$\tilde{n}_c \tilde{n}_e H^x = n_a n_g \left( \sigma_b^x + \phi^{-3/2} n_b \right) + \phi^{1/2} \tilde{n}_a \tilde{n}_g n_b - \phi^{-1/2} \left( \tilde{n}_a n_g + n_a \tilde{n}_g \right) n_b.$$

When $c = 1$ and $e = \tau$ in (E3), it is enforced that $b = d$, and we get the Hamiltonian

$$\tilde{n}_c n_e H^x = -\phi^{-1/2} \tilde{n}_a \tilde{n}_g n_b n_f + \tilde{n}_a n_g n_b \left( \sigma_f^x + \phi^{-3/2} n_f \right) + n_a \tilde{n}_g n_f \left( \sigma_b^x + \phi^{-3/2} n_b \right)$$
$$+ n_a n_g \left( \phi^{-1/2} \left( \sigma_b^+ \sigma_f^- + \sigma_b^- \sigma_f^+ \right) - \phi^{-1} \left( \sigma_b^x n_f + n_b \sigma_f^x \right) - \phi^{-5/2} n_b n_f \right)$$

For $c = \tau$ and $e = 1$, it is enforced that $d = f$, and the Hamiltonian is

$$n_c \tilde{n}_e H^x = -\phi^{-1/2} \tilde{n}_a \tilde{n}_g n_b n_d + \tilde{n}_a n_g \left( R_\tau n_b \sigma_d^+ + R_\tau^* n_b \sigma_d^- + \phi^{-3/2} n_b n_d \right) + n_a \tilde{n}_g \left( R_\tau \sigma_b^- n_d + R_\tau^* \sigma_b^+ n_d + \phi^{-3/2} n_b n_d \right)$$
$$+ n_a n_g \left( R_1 \phi^{-1/2} \sigma_b^- \sigma_d^+ + R_1^* \phi^{-1/2} \sigma_b^+ \sigma_d^- - R_\tau^* \phi^{-1} (\sigma_b^+ n_d + n_b \sigma_b^-) - R_\tau \phi^{-1} (\sigma_b^- n_d + n_b \sigma_d^+) - \phi^{-5/2} n_b n_d \right).$$

Lastly, the Hamiltonian for $c = e = \tau$ is

$$n_c n_e H = \tilde{n}_a \tilde{n}_g \left( R_\tau \sigma_d^+ + R_\tau^* \sigma_d^- + \phi^{-3/2} n_d \right) n_b n_f + \tilde{n}_a n_g \left( R_\tau \phi^{-1/2} n_b \sigma_d^+ \sigma_f^- + R_\tau^* \phi^{-1/2} n_b \sigma_d^- \sigma_f^+ - \phi^{-1} n_b n_d \sigma_f^x \right.$$
$$\left. - \phi^{-5/2} n_b n_d n_f - R_\tau \phi^{-1} \sigma_d^+ n_b n_f - R_\tau^* \phi^{-1} \sigma_d^- n_b n_f \right) + n_a \tilde{n}_g \left( R_1 \phi^{-1/2} \sigma_b^- \sigma_d^+ n_f + R_1^* \phi^{-1/2} \sigma_b^+ \sigma_d^- n_f \right.$$
$$\left. - \phi^{-5/2} n_b n_d n_f - R_\tau \phi^{-1} (\sigma_b^- n_d + n_b \sigma_d^+) n_f - R_\tau^* \phi^{-1} (\sigma_b^+ n_d + n_b \sigma_d^-) n_f \right) + n_a n_g \left( R_1^* \phi^{-1} \sigma_b^+ \sigma_d^- \sigma_f^+ \right.$$
$$+ R_1 \phi^{-1} \sigma_b^- \sigma_d^+ \sigma_f^- + \phi^{-2} n_b n_d \sigma_f^x + \phi^{-7/2} n_b n_d n_f + R_\tau^* \phi^{-1/2} (\sigma_b^+ \sigma_f^- + \sigma_b^+ \sigma_f^-) n_d$$
$$+ R_\tau \phi^{-1/2} (\sigma_b^- \sigma_f^- + \sigma_b^- \sigma_f^+) n_d + R_\tau^* \phi^{-2} (\sigma_b^+ n_d + n_b \sigma_d^-) n_f + R_\tau \phi^{-2} (\sigma_b^- n_d + n_b \sigma_d^+) n_f$$
$$\left. - R_1^* \phi^{-3/2} \sigma_b^+ \sigma_d^- - R_1 \phi^{-3/2} \sigma_b^- \sigma_d^+ - R_\tau^* \phi^{-3/2} \sigma_d^- \sigma_f^+ - R_\tau \phi^{-3/2} \sigma_d^+ \sigma_f^- \right).$$

It can be checked numerically that the x-link Hamiltonian has the same eigenvalues as the z-link Hamiltonian, but larger degeneracies. Finally, the y-link Hamiltonian is

$$
\cdots = \sum_{b',c',d',f'} \sqrt{\frac{d_{b'}d_{c'}d_{d'}d_{f'}}{d_x^4 d_b d_c d_d d_f}} \quad \cdots
$$

(E4)

$$
= \sum_{b',c',d',f'} (R_{c'}^{ch})^* [F_a^{b'\tau\tau}]_{b\tau} [F_{d'}^{c\tau b}]_{c'b'} [F_b^{d'\tau c}]_{dc'} [F_e^{f'\tau d}]_{fd'} [F_g^{\tau\tau f}]_{\tau f'} \sqrt{d_\tau} \quad \cdots
$$

As discussed in Section VI, y-link and x-link are related by combined parity symmetry and time-reversal acting as complex conjugation. This implies that the matrix elements of the y-link Hamiltonian are equal to the complex conjugated matrix elements of the x-link Hamiltonian,

$$
H^y_{abcdefg} = (H^x_{abcdefg})^*.
$$

### Appendix F: Commuting projector fusion surface models and their relation to (enriched) string-nets

Here we discuss the connection between fusion surface models with a commuting projector Hamiltonian and the string-net models previously studied in the literature [22, 35–40]. A summary of this comparison is provided in Table I.

| model | input | topological order | time reversal | chiral central charge |
|---|---|---|---|---|
| Levin-Wen string-net [22] | UMTC $\mathcal{C}$ | $\mathcal{Z}(\mathcal{C}) = \bar{\mathcal{C}} \boxtimes \mathcal{C}$ | yes | $c_- = 0$ |
| fusion surface model, $H = -\sum_p B_p$ | UMTC $\mathcal{B}$, $\rho = 0$ | | | |
| generalized string-net [35] | UFC $\mathcal{C}$ | $\mathcal{Z}(\mathcal{C})$ | no | $c_- = 0$ |
| fusion surface model, $H = -\sum_p B_p$ | ($G$-crossed) braided, not modular UFC $\mathcal{B}$, $\rho = 0$ | | | |
| symmetry-enriched string-net [36–38] | $G$-extension $\mathcal{D}$ of UFC $\mathcal{C}$ | $\mathcal{Z}(\mathcal{C})$, symmetry $G$ | no | $c_- = 0$ |
| fusion surface model, $H = -\sum_p B_p$ | $G$-graded multifusion 1-cat, $\rho = 0 = \oplus_{i=1}^n 0_i$ | | | |
| enriched string-net [39, 40] | $\mathcal{A}$-enriched UFC $(\mathcal{X}, F)$ with $F: \mathcal{A} \to \mathcal{Z}(\mathcal{X})$ | $\mathcal{Z}^{\mathcal{A}}(\mathcal{X})$ | no | $c_- \neq 0$ |
| fusion surface model, $H = -\sum_p B_p$ | ($G$-crossed) UBFC $\mathcal{B}$, $\rho \neq 0$ | $\mathcal{Z}^{\bar{\mathcal{C}}}(\mathcal{C}) = \mathcal{C}$ | no | $c_- \neq 0$ |

TABLE I. Comparison between different string-net models with commuting projector Hamiltonians. The equivalent fusion surface model is written below in the blue highlighted rows.

In this section, the fusion surface Hamiltonian is defined as $H = -\sum_p B_p$, where $B_p$ is the commuting projector specified in (8),

$$
B_p = \sum_{b \in \mathcal{B}} \frac{d_b}{D} B_p^{(b)} \quad \text{with } B_p^{(b)} : \quad \cdots \rightarrow \cdots
$$

(F1)

When all vertical legs are labeled by the identity object $\rho = 0$ and the input category is a UMTC $\mathcal{B}$, the fusion surface model with the projector Hamiltonian (F1) reduces to a Levin-Wen string-net [22] with quantum double topological order $\mathcal{Z}(\mathcal{B}) = \mathcal{B} \boxtimes \bar{\mathcal{B}}$, as discussed in Section 5.3 in Inamura and Ohmori [1]. When $\rho = 0$ and the input unitary ($G$-crossed) braided fusion category is not modular, this fusion surface model reduces to a generalized string-net [35]. The original string-net construction [22] assumed isotropy on the plane and the sphere, meaning that the string-net fusion diagrams must be invariant under bendings, as well as under 2-fold rotations and reflections of the tetrahedron depicted below:

$$
\Phi\left( \cdots \right) = \Phi\left( \cdots \right) = \Phi\left( \cdots \right)
$$

(F2)

Here $\Phi(\cdot)$ denotes the evaluation of the diagram, as explained in Section II. Some fusion categories, e.g. the $\mathbb{Z}_3$ Tambara-Yamagami category, violate the conditions (F2), leading to "generalized string-nets" [35]. Such models realize topological order characterized by the Drinfeld centre $\mathcal{Z}(\mathcal{C})$ of the input fusion category $\mathcal{C}$. Notably, they can realize topological orders which are not simply quantum doubles $\mathcal{C} \boxtimes \bar{\mathcal{C}}$. Although generalized string-net models can break time-reversal symmetry, they still maintain a gapped boundary and zero chiral central charge.

The ground state of a generalized string-net satisfies $B_p |\Phi_{\mathrm{GS}}\rangle = 1$ on all plaquettes. It is unique on a disk geometry. Anyonic excitations are created by terminating "string operators". When acting on the ground states, string operators are path-independent. This ensures independence under elementary deformations such as:

$$\left\langle \ \alpha \diagup\!\!\!\!\!\curlyvee \ \Big| \Phi_{\mathrm{GS}} \right\rangle = \left\langle \ \alpha \diagdown\!\!\!\!\!\curlyvee \ \Big| \Phi_{\mathrm{GS}} \right\rangle$$

They thus are labeled by objects in the Drinfeld center of the input fusion category. We draw them below the string-net, following the convention in Lin *et al.* [35]. A two-anyon state is then created by e.g.

$$\alpha \in \mathcal{Z}(\mathcal{C})$$

Because of the commuting projector Hamiltonian, each anyon has a finite gap $\Delta \geq 1$ over the ground state.

String-net models enriched with a symmetry $G$ have been studied in [36–38]. These can be represented by fusion surface models constructed from $G$-graded multifusion 1-categories, as discussed in Section 5.3 in Inamura and Ohmori [1]. In a multifusion 1-category, the tensor unit is no longer a simple object, but decomposes into a sum of simple objects $0 = \oplus_{i=1}^n 0_i$. This induces a grading of the multifusion category as explained in Chang *et al.* [36].

Enriched string-nets [39, 40] are yet another generalization (different from the symmetry-enriched string-nets mentioned above). A special type of them, called self-enriched string-nets in Huston *et al.* [39], seems closely related to commuting projector fusion surface models constructed from a ($G$-crossed) braided UFC with a nontrivial object $\rho \neq 0$ on the vertical legs. The enriched string-nets live on the boundary of a 3+1d Walker-Wang model [81] built from the UMTC $\mathcal{A}$. The invertible Walker-Wang bulk theory $\mathcal{A}$ represents an anomaly of the 2+1d boundary theory. The resulting commuting projector model on the boundary can realize chiral topological orders which are not accessible to anomaly-free commuting projector string-nets. Such chiral topological orders cannot be written as the Drinfeld center of any unitary fusion category. The local Hamiltonian of the 2+1d enriched string-net has the graphical representation

$$H_p : \quad \substack{b \in \mathcal{X}/\mathcal{A} \ | \ a \in \mathcal{A}} \ \boxed{\phantom{xx}} \quad \rightarrow \quad \sum_{x \in \mathcal{X}/\mathcal{A}} \frac{d_x}{D} \ \boxed{\ x \ } \tag{F3}$$

The vertical green dotted lines are labeled by objects in $\mathcal{A}$ and connect the string-net drawn in black to the Walker-Wang bulk. The construction requires the existence of a braided unitary tensor functor $F : \mathcal{A} \rightarrow \mathcal{Z}(\mathcal{X})$ that maps objects in $\mathcal{A}$ to objects in the Drinfeld center of $\mathcal{X}$, so that the composite $\mathcal{A} \rightarrow \mathcal{Z}(\mathcal{X}) \rightarrow \mathcal{X}$ is faithful. As a result, the unitary fusion category $\mathcal{X}$ decomposes into a disjoint union $\{a, b, \dots\} \bigsqcup \{x, y, \dots\}$ of simple objects $a, b, \dots \in \mathcal{A}$ and $x, y, \dots \in \mathcal{X}/\mathcal{A}$. The black planar edges in (F3) are labeled by objects $x, y, \dots \in \mathcal{X}/\mathcal{A}$. Huston *et al.* [39] argue from a physical perspective, and Green *et al.* [40] prove, that the enriched string-net model realizes topological order characterized by the enriched center $\mathcal{Z}^{\mathcal{A}}(\mathcal{X})$. The Drinfeld center of $\mathcal{X}$ can be decomposed as $\mathcal{Z}(\mathcal{X}) = \mathcal{Z}^{\mathcal{A}}(\mathcal{X}) \boxtimes \mathcal{A}$. The enriched center $\mathcal{Z}^{\mathcal{A}}(\mathcal{X})$ contains those anyons in $\mathcal{Z}(\mathcal{X})$ that braid trivially with $\mathcal{A}$. The physical argument is that the anyonic string operators have to braid trivially with the green dotted legs labeled by objects in $\mathcal{A}$ to preserve path-independence,

$$\left\langle \substack{b \in \mathcal{X}/\mathcal{A} \ | a \in \mathcal{A}} \ \Big| \Phi_{\mathrm{GS}} \right\rangle_{\alpha \in \mathcal{Z}(\mathcal{X})} = \left\langle \ \Big| \Phi_{\mathrm{GS}} \right\rangle \Rightarrow \alpha \in \mathcal{Z}^{\mathcal{A}}(\mathcal{X})$$

Apart from the slightly different geometry, the commuting projector fusion surface model (F1) appears to be a special case of the enriched string-net model (F3) with $\mathcal{X} = \mathcal{C}$ and $\mathcal{A} = \bar{\mathcal{C}}$, resulting in chiral topological order $\mathcal{Z}^{\bar{\mathcal{C}}}(\mathcal{C}) = \mathcal{C}$.

**Appendix G: Unitary mapping of the $J_x$-$J_z$ chain to the Ising anyon chain with twisted boundary conditions**

Here we work out the unitary transformation described in (15), (13), (14) for the Ising input category. The terms in the $J_x$-$J_z$ chain have the form

so the Hamiltonian for a $L = 4$ chain is

$$H = -J_y Y_1 X_2 - J_z Z_2 Z_3 - J_y Y_3 X_4 - J_z Z_4 Z_1. \tag{G1}$$

The Hamiltonian (G1) shares the same bond algebra as the periodic $L = 2$ Ising chain $X_1 + Z_1 Z_2 + X_2 + Z_2 Z_1$, and so has the same spectrum. Note, however that it acts on four qubits, and so each level must have degeneracies. Indeed, the Ising chain has no symmetries beyond the usual spin-flip symmetry generated by $X_1 X_2$. The one-form symmetries of (G1) yield three $\mathbb{Z}_2$ conserved charges, namely $Z_1 Z_2$, $Z_3 Z_4$ and $Y_1 X_2 Y_3 X_4$.

We apply the unitary transformation (13) to move one of the additional $\sigma$-legs to the right:

The Hamiltonian (G1) therefore transforms to

$$U_2 H U_2^\dagger = J_y Y_1 Z_2 + J_z X_2 Z_3 + J_y Y_3 X_4 + J_z Z_4 Z_1 \quad \text{with } U_2 = \frac{1}{\sqrt{2}} \begin{pmatrix} 1 & 1 \\ 1 & -1 \end{pmatrix} \tag{G2}$$

where here and below the matrices are written in the $Z$-diagonal basis on the corresponding site(s). The same $\sigma$-leg is moved to the right once more,

This transforms the Hamiltonian (G2) to

$$U_{23} U_2 H U_2^\dagger U_{23}^\dagger = J_y Y_1 Z_2 + J_z X_2 + J_y Z_2 Y_3 X_4 + J_z Z_4 Z_1 \quad \text{with } U_{23} = \begin{pmatrix} 1 & & & \\ & 1 & & \\ & & 1 & \\ & & & -1 \end{pmatrix}. \tag{G3}$$

Next, the other $\sigma$-leg is moved to the right,

transforming the Hamiltonian (G3) to

$$U_{12} U_{23} U_2 H U_2^\dagger U_{23}^\dagger U_{12}^\dagger = J_y X_1 + J_z Z_1 Y_2 + J_y Z_2 Y_3 X_4 + J_z Z_4 Z_1 \quad \text{with } U_{12} = e^{i\pi/8} \begin{pmatrix} 1 & & & \\ & i & & \\ & & i & \\ & & & 1 \end{pmatrix}. \tag{G4}$$

This $\sigma$-leg is moved to the right once more,

The transforms the Hamiltonian (14) to

$$U_2 U_{12} U_{23} U_2 H U_2^\dagger U_{23}^\dagger U_{12}^\dagger U_2^\dagger = J_y X_1 + J_z Z_1 Z_2 + J_y Y_2 Y_3 X_4 + Z_4 Z_1 \quad \text{with } U_2 = e^{i\pi/8} \begin{pmatrix} 1 & i \\ i & 1 \end{pmatrix}. \tag{G5}$$

The spectrum of course remains the same as that of the $L=2$ Ising chain, and the three $\mathbb{Z}_2$ symmetries are generated by $Y_3$, $Z_2 Z_4$, and $X_1 X_2 Y_4$. As demonstrated in the main text, the extras are the remnants of the plaquette 1-form symmetries.

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
