# Peer review of "Generalizations of Kitaev's honeycomb model from braided fusion categories"

_SciPost Physics_

## Round 1 · Referee Report · Anonymous (Referee 1) · 2025-4-3

Strengths

  • conceptual connections between fusion categories, generalizations of the Kitaev honeycomb model and anyon chains
  • points to the importance of non-invertible symmetries
  • technical finesse & mathematical rigor

Weaknesses

  • unfortunately, the exact nature of the "B-phase" of the Fibonacci generalization of the Kitaev model could not be determined
  • numerical results could be presented in an improved fashion

Report

The paper at hand presents a systematic approach to construct fusion surface models, whose principal input is a braided fusion 1-category. Notably, if this category is the Ising fusion category one obtains Kitaev’s spin-1/2 honeycomb model, while starting from other fusion categories will lead to generalizations of the Kitaev honeycomb model, with the authors discussing both Z_n generalizations as well as the quite interesting scenario obtained when starting from the Fibonacci fusion category.

On a technical level, the manuscript harbors a number of gems. For one, the two principal phases of the Kitaev honeycomb model (and its generalizations) are derived from the fusion categories in an elegant manner. The gapped “A” phase is understood as a quantum double model/Drinfeld centre of the input category, which is pretty straightforward. The gapless “B” phase of the Kitaev model is approached in quite an ingenious way — starting from the quasi-1D limit (Jx=Jy, Jz=0) where the 2D systems decomposes into decoupled anyon chains. The latter are well studied and known to exhibit a non-invertible “topological” symmetry, a generalization of Kramers-Wannier duality, that is crucial in protecting the criticality of the anyon chain. The presence of this symmetry, then limits the types of couplings allowed when moving from this special point towards the “B-phase” and the fully frustrated point (Jx=Jy=Jz) in the middle of the triangular phase diagram. This is all done with great technical finesse, including a discussion of coupled-wires constructions, the role of time-reversal symmetry, and an excursion to twist defects. Some elementary numerical simulation results complement the thorough analysis.

On a conceptual level, the paper makes connections to a number of earlier works in mathematical physics, statistical physics, and quantum topology with a drift towards the senior author’s earlier work. This is particularly true for the discussion of the effect of weakly coupling anyon chains following Eq. (43). To my knowledge, the first discussion that upon coupling two anyone chains the “internal” left/right moving edges modes couple and gap out, while the “outer” ones survive, motivating the formation of a chiral liquid has been presented in Phys. Rev. Lett. 103, 070401 (2009) under the heading “Collective States of Interacting Anyons, Edge States, and the Nucleation of Topological Liquids” (see, in particular, Fig. 2 in that paper), and later discussed in more detail, with numerical support, in Phys. Rev. B 83, 134439 (2011) and New J. Phys. 13, 045014 (2011), much prior to Ref. [42] (from 2014) and Ref. [77] (from 2020). This discussion / referencing should be revised.

In summary, this is a beautiful paper that makes very nice connections between fusion categories, generalizations of the Kitaev honeycomb model and anyon chains, while pointing to the importance of non-invertible symmetries. It should be published with some minor modifcations.

Minor remarks:

  • The notion of “ferromagnetic” coupling used in this manuscript, differs from the notion in Ref. [15] as well as subsequent work by the Santa Barbara group in Phys. Rev. Lett. 103, 070401 (2009) [mentioned above]. It would be good to be either consistent here, or to add a footnote explaining the distinct choices in the literature.

  • Fig. 7 appears to make the point that the data follows a 1/Jz behavior. Why not plot the data as a function of 1/Jz then? This would give an almost perfect straight-line behavior of the data.

Requested changes

see report

Recommendation

Ask for minor revision

  • validity: top
  • significance: high
  • originality: high
  • clarity: good
  • formatting: good
  • grammar: excellent

Author:  Luisa Eck  on 2025-05-07  [id 5451]

(in reply to Report 2 on 2025-04-03)

We would like to thank the referee for their constructive feedback and helpful comments, which we address below:

On a conceptual level, the paper makes connections to a number of earlier works in mathematical physics, statistical physics, and quantum topology with a drift towards the senior author’s earlier work. This is particularly true for the discussion of the effect of weakly coupling anyon chains following Eq. (43). To my knowledge, the first discussion that upon coupling two anyone chains the “internal” left/right moving edges modes couple and gap out, while the “outer” ones survive, motivating the formation of a chiral liquid has been presented in Phys. Rev. Lett. 103, 070401 (2009) under the heading “Collective States of Interacting Anyons, Edge States, and the Nucleation of Topological Liquids” (see, in particular, Fig. 2 in that paper), and later discussed in more detail, with numerical support, in Phys. Rev. B 83, 134439 (2011) and New J. Phys. 13, 045014 (2011), much prior to Ref. [42] (from 2014) and Ref. [77] (from 2020). This discussion / referencing should be revised.

Thank you for highlighting these earlier works. We have included the suggested references in the updated discussion.

The notion of “ferromagnetic” coupling used in this manuscript, differs from the notion in Ref. [15] as well as subsequent work by the Santa Barbara group in Phys. Rev. Lett. 103, 070401 (2009) [mentioned above]. It would be good to be either consistent here, or to add a footnote explaining the distinct choices in the literature.

Thanks for pointing this out, we now refer to this coupling as “antiferromagnetic” in line with the conventions in the earlier literature and added a remark that this means favoring the singlet fusion channel.

Fig. 7 appears to make the point that the data follows a 1/Jz behavior. Why not plot the data as a function of 1/Jz then? This would give an almost perfect straight-line behavior of the data.

We agree that plotting the data as a function of $1/J_z$​ makes the scaling behavior clearer and have updated Fig. 7 accordingly.

---

## Round 1 · Referee Report · Anonymous (Referee 2) · 2025-4-6

Strengths

  1. The paper develops new tools to the field of studying 2+1 dimensional phases of matter. Specifically, it discusses how a class of models known as fusion surface models, introduced previously in the literature, can be used to construct chiral topological phases in 2+1 dimensions in a manner that makes their categorical symmetries manifest. It also opens the possibility that some of these models may harbor phase transitions, giving a new way to access 2+1 dimensional conformal field theories and potentially to study the effects of their generalized symmetries. This work thus sets the stage for further studies on the role of generalized symmetries in a variety of models, including chiral topological orders but also, more importantly, potentially in conformal field theories, where their role is much less well-understood.
  2. The examples discussed are concrete and given in sufficient detail that readers with some background in the area will be able to easily follow exactly what the Hamiltonians described are.
  3. To the best of my ability to judge, the paper is comprehensive in its references of prior literature in the area, so that it is clear what the new contributions of this paper are, and how they are related to prior results.
  4. The paper is clearly written.

Weaknesses

  1. The paper does show some numerical results, but these are definitely limited. I view this more as a justification of further work in this area than a flaw in the current paper.
  2. This is a paper that will be of high interest to the specific community of people working on strongly interacting phases and how to constrain them using generalized symmetries. However, it is quite technical and may not resonate outside of this community.

Report

I really appreciated the perspective that this work gives on the models of Inamura and Ohmori. By restricting to a specific set of examples and discussing the resulting Hamiltonians and their various limits explicitly, the work makes apparent many features that are not at all obvious in Ref. [1]. The new contributions of this work are: (1) To definitively demonstrate that some of these models can harbor chiral topological order; (2) to explain how the 2 dimensional models are related to well-studied 1-dimensional "anyon chain" models, which are known to have the same generalized symmetries. In particular the authors discuss the limit of weakly coupled chains, and what can (and cannot) be deduced about the resulting phases from perturbation theory and symmetry analysis. Further, (3) the authors introduce several new explicit models in this class, including Z_3 and Fibonacci fusion surface models, whose phase diagrams may be of interest for further exploration.

Specific comments:
In the intro, the authors say, "By design, they possess mutually commuting local symmetries. The existence of such anomalous 1-form symmetries makes them promis ing candidates for various topologically ordered phases."

I think the authors mean that they possess mutually commuting 1-form symmetries. (Local symmetry could be interpreted as gauge symmetry here). The sentence also suggests that there is a relationship between the fact that the symmetries commute, and that they are anomalous.

In section II, are you implicitly assuming that the fusion category has a trivial Frobenius-Schur indicator? (I think if you had a non-trivial FS indicator, in general you would need to have some extra information about bivalent vertices, as in e.g. Ref. 35 and https://journals.aps.org/prb/abstract/10.1103/PhysRevB.89.195130 . )

The discussion around Eq. 9 would be easier to follow if the J_z term was introduced explicitly earlier (and its difference with the J_x and J_y terms highlighted).

On p. 6, the tetrahedral symbols are discussed without being introduced. If I understand this description correctly, these can be defined using a simple diagram. By the symmetries of the F's, do the authors mean essentially invariance under rotations of the tetrahedron? Again, if this is correct, it can be stated more explicitly with minimal addition to the text.

Requested changes

  1. Refs. 5 and 6 have been the subject of some debate in the experimental community. See, for example, https://www.nature.com/articles/s41563-022-01397-w . Please rephrase this sentence and add the linked reference (which does not support the claim that the edge modes are fermionic).

  2. P. 5, Please clarify the sentence: “Mathematically, the modular tensor category describing the ensuing topological order takes the form B or B ⊠ C, where B is the input category and C denotes another cat- egory describing emergent anyons”. It is not clear to me whether the authors here are describing the 1D case of quantum chains, or are referring to their 2D models.

  3. On p. 10, “The dashed line represents the branch cut, which disrupts the vortex flavor pattern. Its position is a gauge choice, hence only the defect sites at its endpoints are physical.” I am confused about what the authors mean in this statement. The defect is an actual physical change in the lattice, and the branch cut in the wave function is associated with the location of the defect. So I would have said that the entire defect is physically observable.

  4. I’m confused about the statement that sigma applies charge conjugation to h when it crosses it from the top. In the picture, we see that crossing under sigma, we go from an upward pointing h line to a downward pointing h^{-1} line. I thought that the charge conjugation symmetry here would correspond to having h go in and h^{-1} come out (with arrows oriented in the same direction). Here the sigma line does not appear to be permuting the anyons in this way. Instead it is applying an isomorphism that associates the h^{-1} line with a downward arrow to an h line with an upward arrow. The authors should give a slightly more detailed definition of what they mean by charge conjugation here.

  5. Below Eq. 30, the authors mention dmrg on an infinite cylinder. Are the DMRG results being referred to here the ones in Fig. 5? Please clarify with a reference.

  6. Regarding the additional terms in Eq. 36 required to stabilize the Fibonacci chiral order: this looks fine-tuned. Can the authors clarify this point? Given that the result is a gapped phase, one might expect that there is a range of these terms that realizes the phase in question. The authors should also clarify the relationship between this and the ensuing discussion of numerical results — which, again, presumably support a chiral Fibonacci phase.

Recommendation

Publish (surpasses expectations and criteria for this Journal; among top 10%)

  • validity: top
  • significance: high
  • originality: good
  • clarity: top
  • formatting: perfect
  • grammar: excellent

Author:  Luisa Eck  on 2025-05-07  [id 5450]

(in reply to Report 1 on 2025-04-06)

We are grateful to the referee for their careful reading and the many insightful comments, which we address point by point below:

Specific comments:

In the intro, the authors say, "By design, they possess mutually commuting local symmetries. The existence of such anomalous 1-form symmetries makes them promising candidates for various topologically ordered phases." I think the authors mean that they possess mutually commuting 1-form symmetries. (Local symmetry could be interpreted as gauge symmetry here). The sentence also suggests that there is a relationship between the fact that the symmetries commute, and that they are anomalous.

By “local symmetries” we were referring specifically to the plaquette operators, which are manifestations of the $\mathbb{Z}_2$ 1-form symmetry on the smallest closed loops. We agree that the phrasing could be misleading, and have revised the sentence.

In section II, are you implicitly assuming that the fusion category has a trivial Frobenius-Schur indicator? (I think if you had a non-trivial FS indicator, in general you would need to have some extra information about bivalent vertices, as in e.g. Ref. 35 and https://journals.aps.org/prb/abstract/10.1103/PhysRevB.89.195130 . )

That is correct, we assume trivial Frobenius-Schur indicators throughout the paper, as stated on page 2.

The discussion around Eq. 9 would be easier to follow if the $J_z$ term was introduced explicitly earlier (and its difference with the $J_x$ and $J_y$ terms highlighted).

The $J_z$ term is first introduced in Figure 1, and we highlight the difference to the $J_x$ and $J_y$ terms on page 4, before discussing the evaluation of the x-link.

On p. 6, the tetrahedral symbols are discussed without being introduced. If I understand this description correctly, these can be defined using a simple diagram. By the symmetries of the F's, do the authors mean essentially invariance under rotations of the tetrahedron? Again, if this is correct, it can be stated more explicitly with minimal addition to the text.

We have added a diagram illustrating the tetrahedral symbol in the revised manuscript to make the presentation more self-contained. It is correct that these are invariant under the symmetries of a tetrahedron, and we refer to pages 15-16 in https://arxiv.org/pdf/2008.08598 for a detailed discussion.

Requested changes

  1. Thank you for bringing this to our attention. We have revised the corresponding sentence in the introduction and added the suggested reference.
  2. We agree that the sentence was ambiguous. The discussion refers to our 2D models. Specifically, we meant that if the 1-form symmetry associated with the input category B is spontaneously broken, the resulting phase can be topologically ordered and described by a UMTC containing B as a subcategory. We have revised the paragraph to make this clear and avoid confusion caused by the shift between 1D and 2D.
  3. We agree with the referee that the entire defect line connecting the twist defects is observable as a change of the lattice geometry, and have changed the corresponding sentence in the manuscript.
  4. Thank you for this observation. In the TY($\mathbb{Z}_N$​) category, charge conjugation sends an abelian object h to h^{-1}. We agree that the original diagram was misleading: the outgoing line labeled by h^{-1} should have the same arrow orientation as the incoming h line. We have corrected the figure.
  5. Yes, the DMRG results mentioned below Eq. (30) correspond to the entanglement spectra shown in Figure 5. We have added a reference to the figure.
  6. Indeed, Eq. (36) describes a fine-tuned $\mathbb{Z}_3$ honeycomb model with interactions carefully chosen to stabilize chiral parafermion topological order. In the case of more generic nearest-neighbor interactions—as in the standard $\mathbb{Z}_3$​ Kitaev honeycomb model—multiple competing relevant CFT operators couple the critical chains, complicating any definitive identification of the resulting phase. Nonetheless, our numerical results provide evidence for a chiral Fibonacci phase even away from the fine-tuned limit, suggesting that this phase may emerge robustly within a broader region of parameter space. We have revised the corresponding paragraph in the manuscript.

---

## Round 1 · Referee Report · Anonymous (Referee 3) · 2025-4-28

Strengths

1 - Interesting models based on braided tensor categories.
2 - Several explicit models are studied

Weaknesses

1 - Numerics not conclusive
2 - Perhaps hard without proper background

Report

Report on 'Generalizations of Kitaev’s honeycomb model from braided fusion categories' by Eck and Fendley.

This is another interesting paper by Eck and Fendley, studying fusion surface models, based on braided fusion categories.

Given the delay, this report is extremely short, and does not do the paper justice.

Clearly, this is an interesting paper, introducing and studying several 2-dimensional 'fusion surface models'. It is well written, but without the proper background, it is presumably quite hard to follow. Note that I do not think that this is a problem. One of the interesting results is that the model studied in sec. V, based on the Z_N Tambara-Yamagami category, gives rise to chiral topological order, without having to add term that explicitly time reversal symmetry. Even though the Hamiltonian breaks time reversal, it is nevertheless interesting that no explicit 'magnetic field' is necessary.

In all, I recommend this paper for publication in Scipost.

I have a few questions the authors could consider.

A few specific questions.

Does (22) have an interesting form when written in terms of a fusion diagram?
Page 16: What are GSV chains?
Page 16: In order to get an idea of the complexity, what are the Hilbertspace dimensions of the systems studied numerically?

Generic question.

The fusion categories used in the construction are by assumption unitary and multiplicity free. Nowhere in the paper, modularity is considered. Nevertheless, I am wondering if modularity of the underlying fusion category plays any role. For instance, do the possible phases of the model depend on whether the fusion category is modular or not? If the answer is no, the modularity does not play any interesting role, one can ask the question, why not?

Recommendation

Publish (easily meets expectations and criteria for this Journal; among top 50%)

  • validity: high
  • significance: high
  • originality: high
  • clarity: high
  • formatting: excellent
  • grammar: perfect

Author:  Luisa Eck  on 2025-05-07  [id 5449]

(in reply to Report 3 on 2025-04-28)

We would like to thank the referee for the positive feedback and insightful questions, which we address below:

Does (22) have an interesting form when written in terms of a fusion diagram?

Yes, Eq. (22) can be naturally interpreted in terms of a fusion diagram. It effectively represents the product of two neighboring link terms in the Hamiltonian, and we will include a corresponding diagram in the revised version to clarify this point.

Page 16: What are GSV chains?

“GSV” refers to Grover-Sheng-Vishwanath, who proposed a chain model of Majorana fermions coupled to Ising spins that realizes a tricritical Ising transition in https://arxiv.org/abs/1301.7449. We clarified this in the revised manuscript.

Page 16: In order to get an idea of the complexity, what are the Hilbert space dimensions of the systems studied numerically?

The Hilbert space dimension of the smaller 2x2 system is 8050, and of the larger 3x2 system 722250 (conserving translation symmetry in x direction).

The fusion categories used in the construction are by assumption unitary and multiplicity free. Nowhere in the paper, modularity is considered. Nevertheless, I am wondering if modularity of the underlying fusion category plays any role. For instance, do the possible phases of the model depend on whether the fusion category is modular or not? If the answer is no, the modularity does not play any interesting role, one can ask the question, why not?

As far as we can tell, modularity of the input fusion category does not play a decisive role in determining the phases. When the input category is modular, the isotropic phase B can exhibit the topological order of the input theory—such as the Ising topological order in Kitaev’s honeycomb model with a magnetic field. However, this is not guaranteed, and the model can also realize topological phases that are larger than the input category.

---

## Editorial Decision

resubmitted